# Orchestration of protein acetylation as a toggle for cellular defense and virus replication

L.A. Murray[1], X. Sheng[1] & I.M. Cristea[1]

Emerging evidence highlights protein acetylation, a prevalent lysine posttranslational modification, as a regulatory mechanism and promising therapeutic target in human viral infections. However, how infections dynamically alter global cellular acetylation or whether viral proteins are acetylated remains virtually unexplored. Here, we establish acetylation as a highly-regulated molecular toggle of protein function integral to the herpesvirus human cytomegalovirus (HCMV) replication. We offer temporal resolution of cellular and viral acetylations. By interrogating dynamic protein acetylation with both protein abundance and subcellular localization, we discover finely tuned spatial acetylations across infection time. We determine that lamin acetylation at the nuclear periphery protects against virus production by inhibiting capsid nuclear egress. Further studies within infectious viral particles identify numerous acetylations, including on the viral transcriptional activator pUL26, which we show represses virus production. Altogether, this study provides specific insights into functions of cellular and viral protein acetylations and a valuable resource of dynamic acetylation events.

[1] Department of Molecular Biology, Princeton University, Lewis Thomas Laboratory, Washington Road, Princeton, NJ 08544, USA. These authors contributed equally: L.A. Murray, X. Sheng. Correspondence and requests for materials should be addressed to I.M.C. (email: icristea@princeton.edu)

Viruses are obligate parasites whose survival depends on their ability to manipulate host cellular functions to promote viral replication and inhibit host defenses. This orchestration of cellular processes is finely tuned across both space and time to facilitate each step of the virus replication cycle. Proteins underlie many of these alterations and their functions are regulated via changes in protein levels, posttranslational modifications (PTMs), interactions, and subcellular localizations. As infection-induced changes in protein functions are linked to numerous human diseases, understanding this dynamic regulation is critical for defining the biology and pathology of virus infections.

Among these protein alterations during infection, accumulating evidence highlights protein lysine acetylation as a key regulatory point in mechanisms of both host antiviral response and virus replication (Fig. 1). For example, the activity of nuclear factor (NF)-κB, a transcriptional activator of cytokine production, is regulated by acetylation of its RelA subunit[1]. The interferon-inducible protein IFI16, a viral DNA sensor, is also regulated via acetylation within its nuclear localization motif to dictate subcellular localization and thus determine its ability to initiate innate immune response upon infection with DNA viruses, including herpes simplex virus 1 (HSV-1) and vaccinia virus[2]. Viruses have also evolved strategies to inhibit acetylation and host antiviral signaling. An example is the acetylation of the tumor suppressor p53, which is required for its antiviral function[3]. Numerous viruses possess mechanisms to block p53-dependent apoptosis by inhibiting its acetylation, including the herpesvirus human cytomegalovirus (HCMV), human T-cell leukemia virus, the small DNA tumor viruses human papillomavirus and adenovirus type 5, and human immunodeficiency virus type-1 (HIV-1)[4–8]. On the other hand, acetylations have been discovered on viral proteins. Acetylation of the HIV-1 protein Tat is critical for its transcriptional activity[9], and influenza A nucleoprotein acetylation was shown to impact viral replication[10].

Despite these findings, protein acetylation remains an understudied aspect of viral infections and of biological phenomena in general. Although initially regarded as a rare cellular event, research in the past decade has established acetylation as a prevalent PTM, present on thousands of proteins in mammalian cells and tissues[11–13]. However, there is limited information about the dynamic regulation of acetylation during biological processes. Therefore, the functions of most acetylations remain unknown even outside the context of infection. Functional studies on certain cellular processes or proteins of interest have demonstrated that acetylation impacts protein functions in multiple ways (Fig. 1). Acetylation contributes to enzyme activity, chromatin structure, protein localization, and protein–protein interactions, and is a known regulator of transcription and metabolism[14,15]. These core cellular processes are required for viral replication, suggesting that acetylation is an important molecular toggle of protein functions during infection. Support for this also comes from the findings that the activity of enzymes that control protein acetylation levels, including histone deacetylases (HDACs), histone acetyltransferases, and sirtuins (SIRTs), impacts viral replication. HDAC inhibitors were shown to induce the reactivation of prominent human viruses from latency, such as HCMV, HSV-1, and HIV-1[16–18]. SIRTs, another family of deacetylases, were found to have antiviral roles against several DNA and RNA viruses, including HCMV, HSV-1, adenovirus, and influenza A[19]. Furthermore, small molecules that activate SIRTs inhibited the replication of HCMV, indicating that the regulation of acetylation

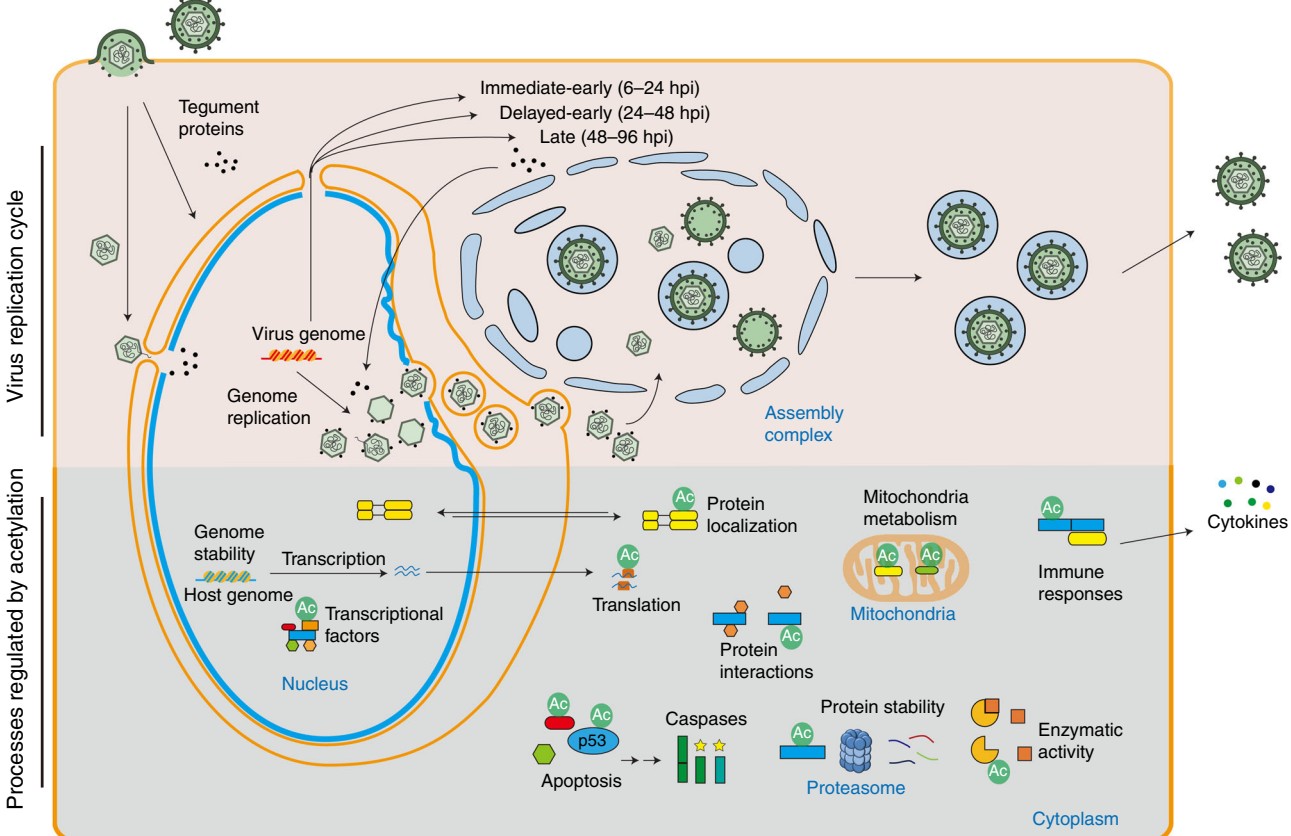

**Fig. 1** Acetylation contributes to diverse protein functions in uninfected and infected cells. Host pathways that are regulated by acetylation in uninfected cells are manipulated by HCMV during its temporal replication cycle

provides a therapeutic tool against infections. However, understanding the contribution of acetylation to host defense or virus replication is limited by the lack of knowledge of its temporal regulation during an infection process.

HCMV provides an important biological and clinical model for studying protein acetylation during infection. This beta-herpesvirus establishes life-long infections and is a widely spread human pathogen, infecting 40–100% of the adult population worldwide. Although HCMV infections are a major cause of virus-induced birth defects and mortality and morbidity in immunocompromised patients, we still lack a vaccine and effective antiviral treatments. HCMV is an ancient virus and, given its long co-evolution with its host, it has acquired numerous mechanisms for altering cellular transcription, metabolism, and defense responses. Underlying these mechanisms are striking changes in the composition and shape of subcellular organelles, which have become hallmarks of different stages of the HCMV replication cycle[20] (Fig. 1). Mitochondrial morphology is disrupted to manipulate cellular metabolism, the nuclear periphery is reshaped to aid virus nuclear egress, and the endoplasmic reticulum (ER) and Golgi help to form a viral assembly complex where new viral particles are made. Acetylation is known to be prominent in the proteomes of each of these subcellular compartments in uninfected cells, yet their functions, temporal regulation, and contributions to infection remain unexplored.

Here, we examine the dynamic protein acetylome during virus infection. By integrating molecular virology, quantitative proteomics, and functional analyses, we characterize spatiotemporal acetylations on cellular and viral proteins throughout the HCMV replication cycle in human fibroblasts. We define the regulation of protein acetylation in the context of both protein abundance and subcellular localization across infection time. We discover that lamin acetylation at the nuclear periphery inhibits lamina disruption and virus capsid egress, thereby restricting virus production. Further studies of purified infectious viral particles reveal prominent acetylations within all virion compartments, and mutagenesis assays show the impact of acetylation on virus replication. Altogether, this study provides temporal resolution for both the human and HCMV acetylomes and establishes acetylation as a highly regulated molecular toggle of protein function during viral infection.

## Results

**Workflow investigating acetylome dynamics during infection**. To investigate cellular and viral protein acetylomes in space and time during HCMV infection, we designed a mass spectrometry (MS)-based proteomics workflow to quantify protein acetylation in conjunction with protein abundance (Fig. 2a). Acetylations on viral proteins were determined both in the context of infection and in isolated infectious viral particles. The cellular acetylome was investigated during the distinct stages of the HCMV replication cycle by collecting primary human fibroblasts at time points representing early (24 hpi), delayed early (48 hpi), and late (72 and 96 hpi) stages of viral infection, in addition to mock (uninfected) cells (Fig. 2a). Changes in site-specific acetylations were determined by enriching acetylated peptides with anti-acetyl lysine antibodies, and analyses were performed in three biological replicates for all time points. The quality of our acetylome data set was determined using several criteria. First, we assessed the specificity, reproducibility, and depth of analysis. Similar to acetylome reports from other biological systems[21,22], we achieved ~ 30.0% acetyl-enrichment, and identified 6180 acetylated human peptides corresponding to 2018 acetylated proteins. Second, these identifications were not limited to abundant proteins. When accounting for protein abundances in human cells (PaxDb),

acetylations in our data set were detected on proteins ranging in abundance from 0.005 to 13,168 parts per million (ppm) (i.e., C4orf47 to GAPDH), including low abundance components of transcriptional regulatory complexes (Fig. 2b, c). Finally, we assessed whether the enriched motifs of acetylated peptides are similar to those previously reported for human cells. As reported for uninfected mammalian systems[12,23], we found glutamate, aspartate, and glycine enriched at the − 1 position, the − 3, and − 4 positions enriched for glutamate, and the +1 to +6 positions enriched for lysine throughout infection (Fig. 2d, Supplementary Fig. 1a–e). Altogether, these assessments indicated that our temporal acetylome data set provides the depth, specificity, and efficiency of enrichment necessary for discovering changes in protein acetylation during the progression of infection.

**Temporal acetylations decorate metabolic and immune proteins**. As changes to protein abundance can influence the observed alterations in protein acetylation, we performed parallel whole-cell proteome analyses throughout the HCMV infection (Supplementary Data set 1, Supplementary Fig. 2a). The reproducibility in the biological triplicates was supported by the coefficient of variation (CV), with 95% of the data having CV ≤ 54%, and the depth of analysis was in line with previous proteome investigation during HCMV infection[24] (Supplementary Fig. 2b–e). Furthermore, the changes in protein abundance matched those expected[20,24], such as the upregulation of interferon response proteins (Fig. 3a). In agreement with the known HCMV-induced increase in flux through metabolic pathways[25], proteins related to oxidative phosphorylation and fatty acid biosynthesis were slightly elevated. Also as expected[24], proteins that regulate the actin cytoskeleton and focal adhesion declined in abundance during infection (Fig. 3a). These observations provide confidence for the use of our proteome data as a normalization tool to define how cellular acetylation changes in time during infection. In addition, this proteome data set provided further confirmation that acetylations were detected on proteins with a range of abundances by using intensity-based absolute quantification (iBAQ) (Supplementary Fig. 1f).

To determine temporal changes in acetylation, we focused on identifications that were consistent between biological replicates at the different time points (Supplementary Fig. 3a). Specifically, we required each acetylated peptide to be present in two out of the three replicates in at least one time point, and to have a %CV < 100 for at least one time point (Supplementary Fig. 3b). Out of the identified 6180 host acetylated peptides, 5756 passed these criteria, corresponding to 1816 acetylated proteins (Supplementary Data set 2, Supplementary Fig. 4a). Hierarchical k-means clustering of quantified acetylated peptides highlighted distinct temporal profiles of either elevated or decreased acetylations during infection (Fig. 3b, right, Supplementary Fig. 5a). Given the striking number of changes in acetylation levels and their specific temporal signatures, our results suggest that protein acetylation provides an important regulatory mechanism during infection. Furthermore, by normalizing acetylation levels to protein abundance, we found that a majority of the identified alterations in site-specific acetylations are independent of protein abundance changes (Fig. 3b). This conclusion is reinforced by the low correlation between the changes in protein abundances and acetyl-peptide abundances (Supplementary Fig. 4b–e).

One exception to this independence from protein abundance is the acetylation of proteins involved in the interferon and innate immune response. These proteins, including the interferon-induced proteins with tetratricopeptide repeats IFIT1, IFIT2, and IFIT3, are upregulated early in HCMV infection[24]. As innate immune response proteins are

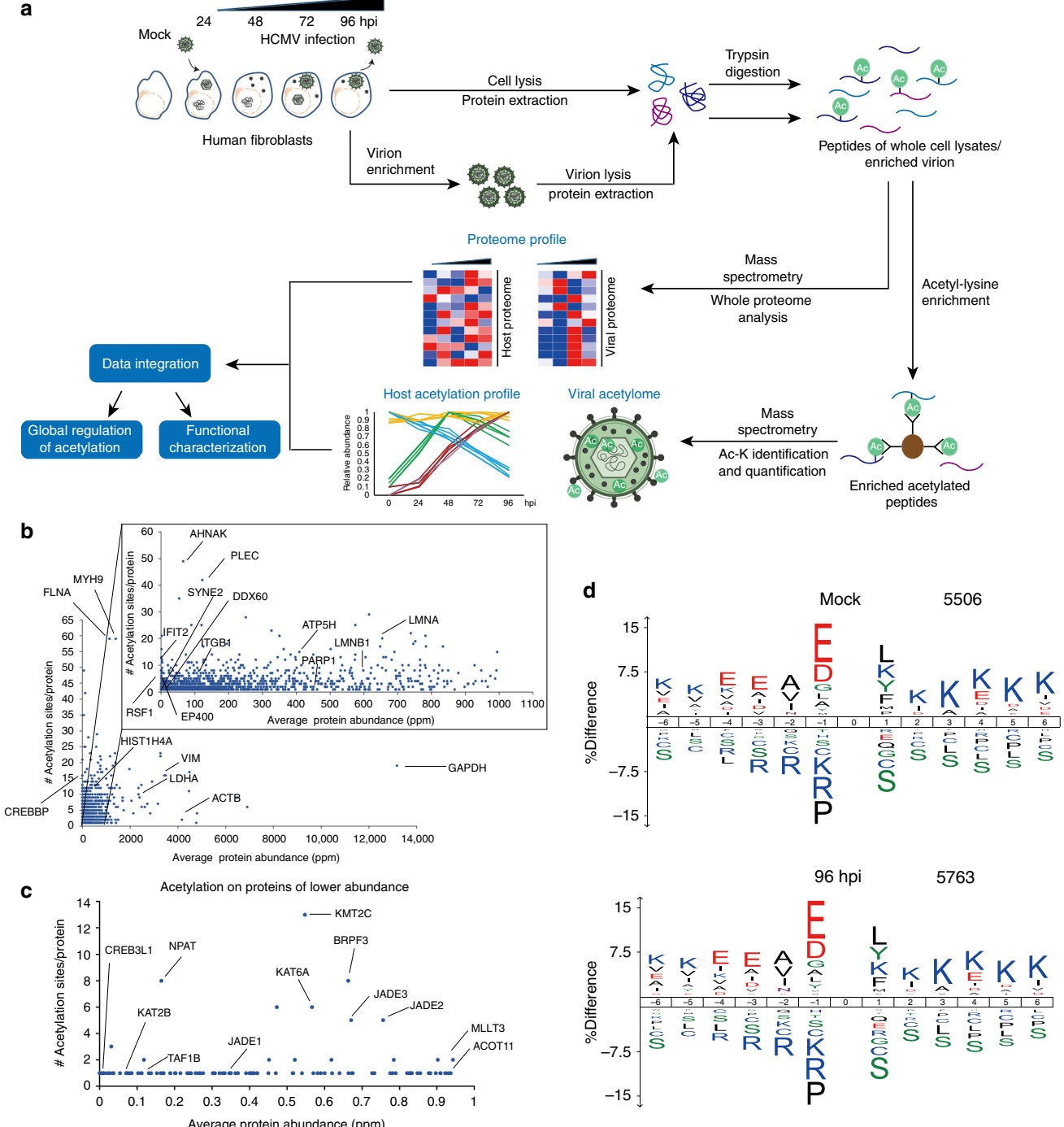

**Fig. 2** Dynamic regulation and function of protein acetylation during HCMV infection. **a** Workflow of anti-acetyl lysine IP and MS from whole-cell lysate and enriched virions. **b** Investigating the relationship between number of acetylation sites and protein abundance. Average protein abundance in parts per million (ppm) sourced from PaxDB; Proteins in this database are scaled in ppm relative to the whole proteome of the reference data set; in this case, the abundances were drawn from the data set: *H. sapiens*–whole organism (integrated). **c** Examples of acetylations on low abundance proteins. **d** Acetylation motifs in uninfected (mock) and infected (96 hpi) cells, analyzed by IceLogo. Numbers of peptides used for motif analyses are indicated. Analysis of all time points is provided in Supplementary Fig. 1

expressed at low levels or not at all in uninfected cells, the lack of detection of many acetylated peptides may represent either a specific role during infection or their presence below the limit of detection in uninfected cells (Fig. 3c). Consequently, for the acetylated peptides detected only in infection (Fig. 3c, red boxes), we visualized the changes on a 0–1 scale. However, for the acetylated peptides that were also detected in uninfected

cells, we normalized their levels to protein abundance. This normalization indicated that the increase in protein abundances for IFIT proteins often outweighs the increase in acetylated peptide abundances (Fig. 3c, Supplementary Fig. 5b–d). The fact that the majority of our identified acetylations on these proteins have not been previously reported (red lines) and were not detected in uninfected cells (red boxes) suggests

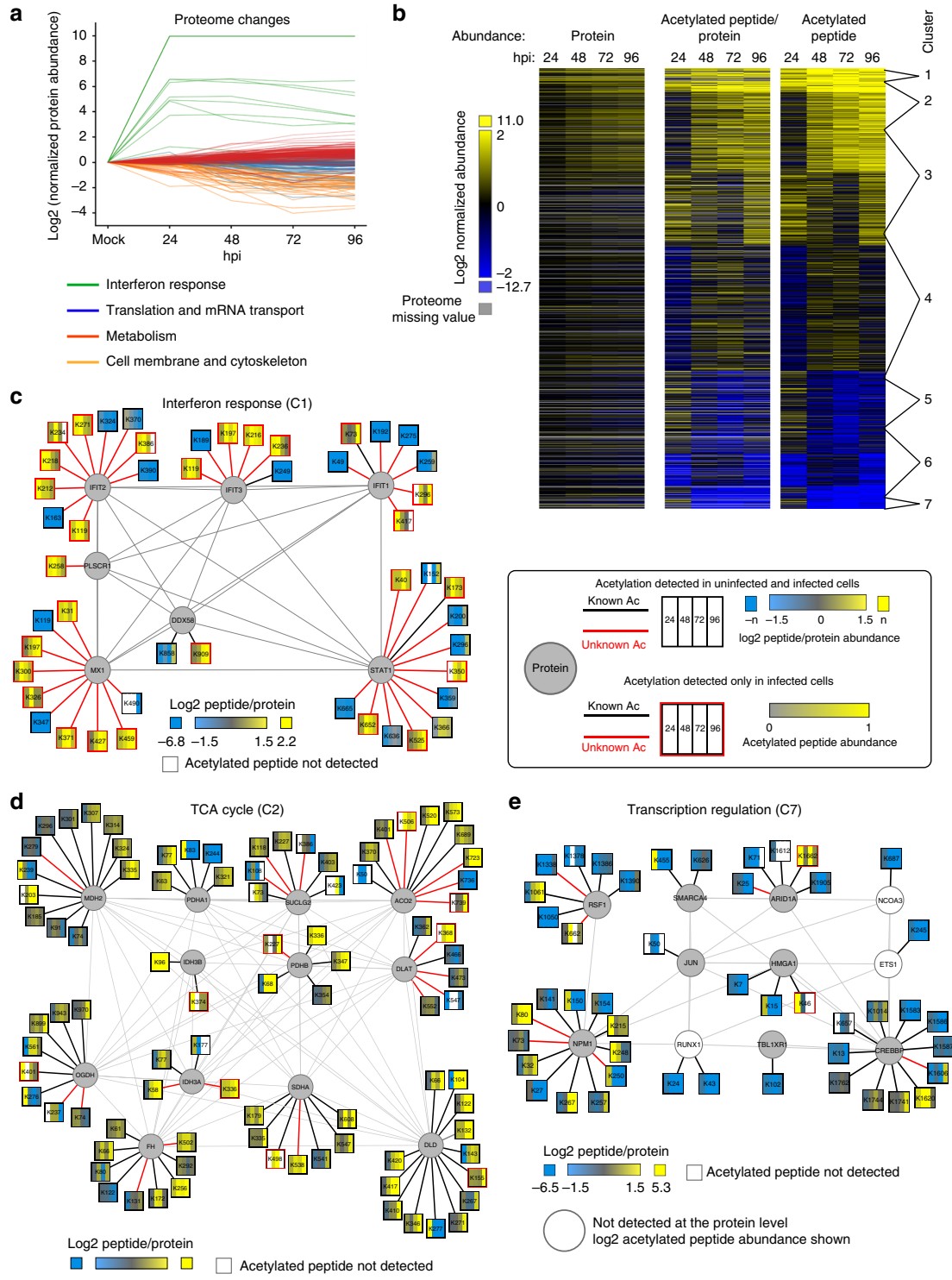

**Fig. 3** Acetylation increases on proteins in immune response and metabolic pathways and decreases on transcriptional regulators. **a** Protein abundances in infection were normalized to mock abundances and plotted as log2 values. Four biological processes relevant to HCMV infection that display distinct protein abundance trends are illustrated. **b** K-means clustering was conducted in C3 ($k = 7$) to cluster changes in peptide acetylation levels at different time points of infection when compared with uninfected cells (right column). Whole-cell protein abundances (left) and peptide acetylation levels normalized to protein abundance (middle) are illustrated. Increased abundance, yellow; decreased abundance, blue; not detected in the proteome, gray. **c–e** Detected acetylation sites on proteins from indicated functional clusters representing site-specific regulation of acetylation during infection. Circles, proteins; squares, acetylation sites; red lines, previously unknown acetylation sites; black lines, known acetylation sites; red boxes, sites only detected during infection with abundances displayed on a 0–1 scale as they could not be normalized to mock samples; white bars, acetylated peptide not detected. **c** interferon response, **d** TCA cycle, **e** transcription regulation

these proteins need to be stimulated (via infection or other forms of interferon induction) for detection.

The temporal acetylome also highlighted the prominence of site-specific regulation during infection, as numerous proteins contained acetylations that were either upregulated or down-regulated at distinct sites. This included proteins that displayed some of the most pronounced increases in acetylation, such as those involved in the TCA cycle (Fig. 3d). For example, DLD, a subunit of multiple metabolic enzyme complexes, has up to a fivefold increase in acetylation at multiple sites during infection. On the other hand, DLAT, a subunit of the pyruvate dehydrogenase complex, shows a 3.5-fold decrease in K547 acetylation as early as 24 hpi, whereas K368 acetylation is not detected until 48 hpi. As acetylations can contribute to both enhancing and repressing the catalytic activity of metabolic enzymes[26], these previously unknown DLAT acetylations may provide nuanced regulation of protein function during infection. HCMV is known to induce an increase in metabolic flux and alterations in mitochondrial morphology[20,25]. Although acetylation has not been studied during HCMV infection, acetylation on metabolic proteins in *Salmonella enterica* was reported to enhance flux through the metabolic pathway in response to different carbon sources[27]. Therefore, acetylation likely provides an additional mechanism by which HCMV infection alters cellular metabolism.

Our data set also pointed to decreased acetylations during infection, including on transcriptional regulatory proteins (Fig. 3e). For example, ARID1A, a component of the DNA-binding barrier-to-autointegration factor SWI/SNF complex, displays a threefold decline in acetylation on K25, K71, and K1905 at 48 hpi, and this decline persists later in infection. ARID1A has been identified as a substrate of the deacetylase HDAC8[28], though the specific regulated site was not determined. The dynamic acetylations identified here may contribute to the regulation of ARID1A activity to impact viral or host gene expression during infection.

Altogether, our study uncovers dynamic protein acetylations that are finely tuned temporally and in a site-specific manner. However, the finding of altered acetylations on proteins known to reside in distinct subcellular organelles, ranging from mitochondrial proteins involved in the TCA cycle and ATP synthesis to nuclear proteins involved in transcriptional regulation and mRNA splicing (Supplementary Fig. 6a–b), further suggests a localization-specific regulation of acetylation during HCMV infection.

**Subcellular organelle-specific acetylation profiles**. Spatial regulation of protein functions is essential during HCMV infection, for example facilitating trafficking, maturation, and release of viral particles. To interrogate dynamic acetylations within different organelles, we next integrated the acetylome and proteome data sets with our knowledge of temporal cellular protein localization at these different stages of HCMV infection[20] (Fig. 4a, Supplementary Data set 3). We previously used live cell microscopy, density fractionation, and quantitative MS to characterize protein localization to specific organelles during infection. This allowed us to predict whether the identified acetylated proteins localize to certain subcellular compartments. In addition, reanalysis of the previous organelle proteome data set[20] also identified a small subset of the acetylated proteins reported here (Supplementary Data set 4), providing confidence to the assigned subcellular localization for these acetylated proteins. Many of these acetylations occur on organelle resident proteins that do not change localization during infection. However, increased and decreased temporal acetylation trends were noticeable within each organelle.

Mitochondrial proteins displayed a striking global increase in acetylations. This points to the likely dynamic regulation of metabolic pathways during infection, in agreement with reports on HCMV-induced changes in cellular metabolism[25]. Although not previously studied during infection, global elevation in mitochondrial acetylation was reported to be detrimental to oxidative and intermediary metabolism[29]. Such mitochondrial dysfunctions have been implicated in metabolic syndrome and cardiac hypertrophy[30,31], diseases linked to HCMV infection. Whether this connection is in part regulated through protein acetylation remains to be investigated. Of note, metabolic pathways display distinctive temporal patterns. For instance, acetylation is increased throughout infection on proteins involved in ATP synthesis and oxidative phosphorylation, whereas those involved in amino acid degradation show delayed increase. These temporal differences may be driven by specific virus requirements or host responses at distinct infection stages. Only a minor subset of mitochondrial proteins displayed decreased acetylation during infection, and these included regulators of mitochondrial architecture, such as MICOS complex subunit MIC25 (CHCHD6), and regulators of membrane potential, such as the voltage-dependent anion-selective channel protein 3. As HCMV was shown to induce disruption of mitochondrial membrane potential[32], fragmentation, and calcium influx from ER[33], our findings point to acetylation as a potential regulator of mitochondrial plasticity during HCMV infection.

In contrast, other subcellular compartments displayed mixed temporal dynamics of acetylations. In the cytosol, in addition to the interferon response proteins, we observed increased acetylations on proteins involved in mRNA stability and ubiquitin ligase activity during late infection. HCMV is known to regulate cellular mRNA levels and ubiquitination-dependent protein degradation to suppress expression of interferon response genes and inflammatory cytokines[34]. Changes in acetylation were also assigned to the ER on proteins involved in protein folding and processing, adding a possible layer of regulation to the known HCMV-induced stimulation of the unfolded protein response[35]. In the nucleus, however, most acetylations declined during infection, including on proteins involved in nucleosome disassembly and transcriptional activation. As the site for viral DNA transcription, replication, and viral capsid assembly, the nucleus is a primary target of HCMV manipulation. As acetylation is often a positive regulator of transcription, these declining trends could contribute to the HCMV-driven repression of host antiviral factors. Notably, a subset of nuclear proteins, including nuclear periphery-associated proteins, displayed increased acetylation during infection.

Our analysis also uncovered acetylations on proteins previously predicted to either have multiple localizations or translocate between subcellular compartments during HCMV infection[20] (Fig. 4b). The possible acetylation of translocating proteins was supported by our reanalysis of the Jean Beltran et al.[20] data set, which identified a common subset of acetylations on proteins predicted to change localization (Supplementary Data set 4). Furthermore, these proteins are also known to have multiple localizations or to translocate outside the context of infection. Among these is myosin 9 (MYH9), which functions in actin remodeling[36] and, in conjunction with MYH10, in nuclear re-positioning. This is of relevance during infection given the report that Golgi-derived MTOC-induced nuclear rotation is critical for maintenance of the HCMV assembly complex (AC)[37]. As MYH9 and MYH10 were reported to traffic to the AC late in infection[20], it remains to be determined whether the identified concomitant increase in acetylation at the previously unreported K1538 site is part of a localization-specific function of MYH9. Other identified alterations in acetylation in our data set also

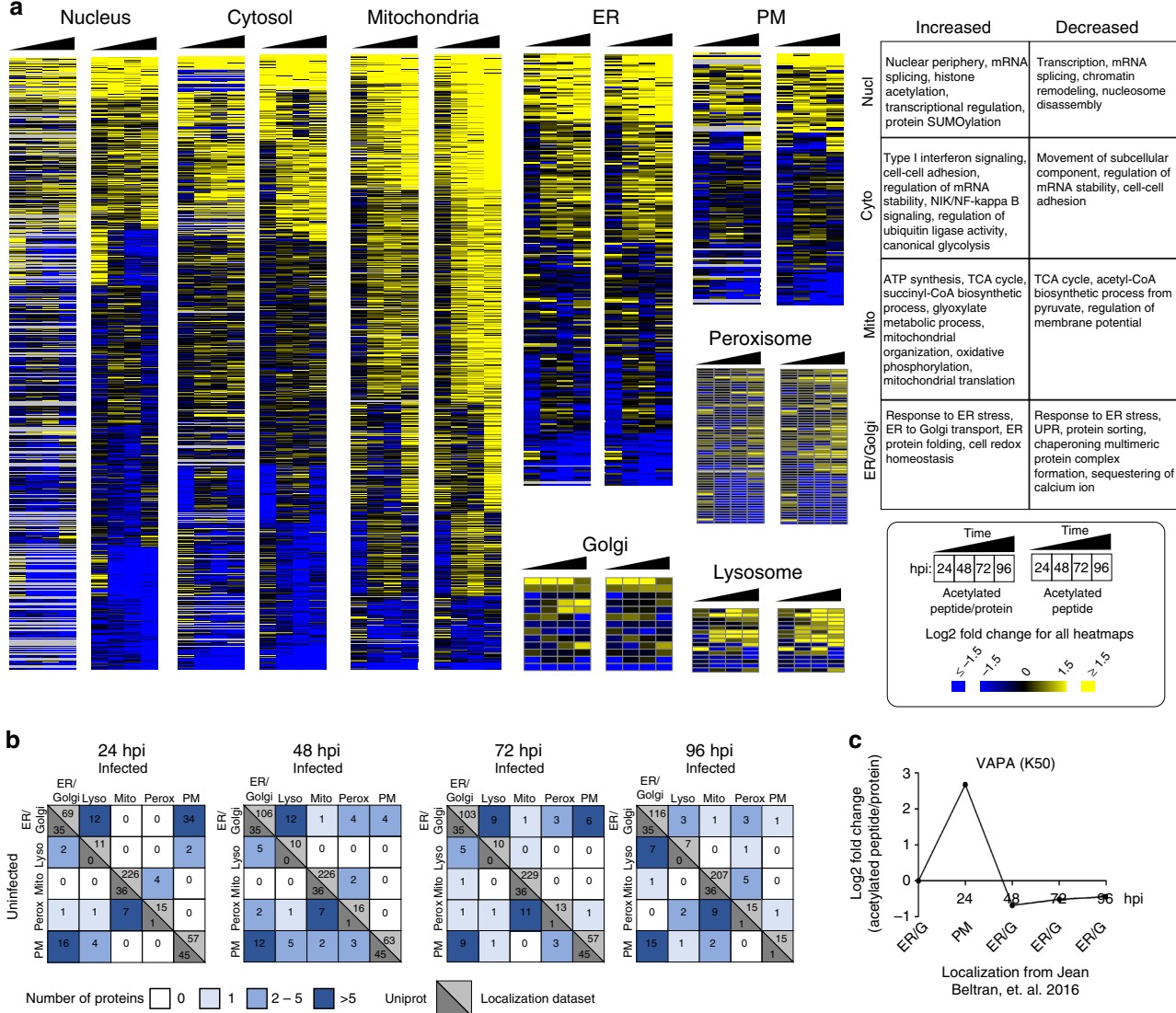

**Fig. 4** Acetylation in the context of temporal subcellular localization and translocation during infection. **a** Heatmaps visualizing temporal changes in peptide acetylation levels (right) and acetylation normalized to protein abundance (left) in the nucleus, cytosol, mitochondria, ER, Golgi, plasma membrane (PM), peroxisome, and lysosome. Representative biological processes overrepresented in clusters of increasing and decreasing acetylations within specific subcellular compartments are shown. **b** Predicted localizations of the identified acetylated proteins in uninfected (mock) cells (vertical) are compared to the localizations of the proteins at 24, 48, 72, and 96 hpi (horizontal) (from Jean Beltran, et al. 2016[20]). Numbers in the upper, light gray triangles along the diagonals indicate non-translocating proteins annotated from[20], and those in the lower, dark gray triangles indicate proteins annotated from Uniprot. Boxes displaced from the diagonal designate predicted putative translocating protein localization at given time points during infection. Numbers of translocating proteins are indicated by color code. As an example, of the proteins assigned to the ER/Golgi in uninfected cells, 34 were assigned as translocated to the PM at 24 hpi. **c** Example of a protein translocating from ER/G to PM at 24 hpi. A change in protein acetylation is also observed at 24 hpi. Peptide acetylation levels normalized to protein abundance are represented in parallel with organelle subcellular localization (assigned from[20]) at each infection time point

temporally coincided with proposed translocation events during HCMV infection. It remains to be seen whether some of these acetylation events can contribute to the observed changes in localization or are involved in localization-specific functions. One possible example is the early increase (24 hpi) in the previously unknown K50 acetylation on the vesicle-associated membrane protein-associated protein A (VAPA) (Fig. 4c). VAPA has roles in endocytic trafficking[38], transport and contacts between the plasma membrane (PM) and the ER[39], and is involved in the stimulation of the HCMV receptor ITGB1[40]. In agreement with its functions, a temporal shuttling of VAPA between the ER/Golgi and the PM was reported during HCMV infection[20] (Fig. 4c).

Altogether, the integration of our acetylome data set with spatial-temporal information of protein localization during HCMV infection provides a perspective of how site-specific, temporal acetylation can be regulated at a subcellular level during virus replication. This can inform future work on the role of acetylation in organelle-specific protein function and possibly altered subcellular localization during infection.

**Lamin B1 acetylation inhibits viral capsid nuclear egress**. Our prediction of spatial acetylation pointed to a cluster of nuclear proteins with elevated acetylation during infection (Supplementary Fig. 7a). This cluster contained nuclear periphery proteins,

including the lamins A/C (LMNA/C) and B (LMNB1 and LMNB2) (Fig. 5a, Supplementary Fig. 7a). Altogether, our data set revealed acetylations on six laminar proteins during infection (Fig. 5b). Previous work revealed that nuclear architecture is dramatically altered late during HCMV infection to facilitate viral capsid egress from the nucleus and the formation of the viral AC[41] (Fig. 1). To make this possible, LMNA/C is disrupted through a phosphorylation-dependent mechanism[42,43]. However, to what extent and how B lamins are also disrupted during HCMV infection remains poorly understood. Among these proteins, LMNB1 had the most pronounced increases in acetylation at several sites (up to six-fold), as well as a decrease at one site (down by seven-fold). Most of these lysine residues are conserved in LMNB1 across vertebrates, whereas K102 and K134 seem to have evolved in mammalian species (Fig. 5c). The sites that had the most pronounced acetylation changes late in infection when compared to uninfected cells or to early infection were K109 (decreased) and K134 and K483 (increased). K109 and K134 are in Coil 1 (Fig. 5d) that mediates homotypic interactions to build the laminar network, whereas K483 is located in the Ig-like fold domain that interacts with other non-lamin proteins[44]. Given the location of these sites in functional regions of LMNB1 and the critical role of the lamina in virus capsid egress, we asked whether LMNB1 acetylation can impact virus production. We generated acetyl-mimic (glutamine) and charge-mimic (arginine) mutants for K109, K134, and K483, and confirmed the correct localization of these mutants at the nuclear periphery (Fig. 5e). We next assessed the effect of these LMNB1 mutants on the amount of infectious extracellular virus produced. A 200-fold reduction in viral titers (measured by IE1 staining) was observed for the K134Q mutant and a more modest three-fold reduction for the K134R mutant when compared to WT LMNB1, but no significant effect was seen for the K109 or K483 mutants (Fig. 5f–h). The impact of the K134Q mutant on virus titers was also confirmed using plaque assay (Supplementary Fig. 7b). The increase in acetylation late in infection, in conjunction with this striking impact on virus titers, led us to propose that LMNB1 K134 acetylation may affect nuclear egress of viral capsids. To start to address this, we measured cell-associated virus. Cells expressing K134Q yielded significantly less infectious cell-associated virus in comparison with cells expressing WT LMNB1, whereas K134R had an intermediate and not-significant phenotype (Fig. 5i). This supports that the reduced virus titers observed for K134Q can be in part driven by impaired capsid nuclear egress. To further address this hypothesis, we used live cell imaging to compare virus capsid localization in human fibroblasts expressing either mCherry-tagged K134Q, K134R, or WT LMNB1 (Fig. 6a, Supplementary Fig. 9). To track viral capsids, we infected these cells with an HCMV strain expressing GFP-tagged pUL32, a viral protein known to directly bind the capsid[43]. At 96 hpi, nuclear capsid egress progressed similarly in cells with K134Q or WT LMNB1, and even more effectively in K134R cells, as confirmed by quantifying nuclear pUL32 signal intensity (Fig. 6b, c). However, at 120 hpi, more viral capsids were accumulated in the nucleus in the presence of the K134Q mutant in comparison with the WT and K134R. 3D reconstruction of pUL32-containing particles suggested increased aggregation of viral capsids within the nucleus in the presence of K134Q (Fig. 6d). Altogether, these results suggest that LMNB1 acetylation can impact lamin regulation during infection. To better understand what drives this, we further investigated lamin morphology in the presence of these mutations late in infection (96 and 120 hpi, Fig. 6e, Supplementary Fig. 10). The cells expressing WT or K134R LMNB1 displayed the known disruption of the nuclear periphery late in infection, with infoldings of the lamina and the formation of a kidney-shape nucleus. The curvature of this kidney shape is

known to accommodate the viral AC (Fig. 1, Fig. 6a). In contrast, the cells expressing K134Q displayed a significant reduction in the disruption of the nuclear periphery, with less infoldings observed (Fig. 6e–f). Kidney-shaped nuclei were visible, but had a less pronounced aspect than in WT and K134R LMNB1 cells (Fig. 6e). These results indicate that K134 LMNB1 acetylation can inhibit the disruption of the nuclear lamina, thereby interfering with viral capsid nuclear egress.

**Functional viral acetylation within infectious particles**. In addition to the numerous identified cellular acetylations, we also discovered acetylated viral proteins (Supplementary Data set 5, Supplementary Fig. 11a, b). Viral protein acetylation is an area currently understudied, and all the identified viral acetylation sites have not been previously reported. Furthermore, these acetylations were present on viral proteins from the different temporal stages of infection, i.e., IE proteins pUL122 and pUL123, DE proteins pUL26, pUL32, pUL34, pUL44, pUL48, pUL50, pUL54, pUL84, and US22, and L proteins pUL31, pUL46, pUL69, pUL83, pUL85, pUL86, and pUL94 (Fig. 7a). Viral protein acetylation increased with infection time, in agreement with the temporal cascade of viral gene expression. When normalizing acetylation levels to protein abundances at the corresponding time points, several trends were noticed. For example, the viral proteins pUL122 and pUL83 were already acetylated early in infection, and their acetylation decreased late in infection. As the tegument protein pUL83 is deposited in the cell immediately upon infection, and pUL122 is an immediate early viral protein, these early acetylations match the temporal protein profiles. Although pUL83 is known to suppress host immune responses, its phosphorylation at S364 was shown to compromise its ability to inhibit the host defense factor IFI16[45]. The identified temporal regulation of its K378 acetylation could further modulate pUL83 functions. Another example is the acetylation on pUL26 K203 within a domain essential for viral replication[46]. This acetylation is prominent at 24 and 48 hpi, decreasing at 72 and 96 hpi. In addition, several acetylations were detected predominantly late in infection, although site-specific modulation was observed. For example, increased late in infection were K413, K439 and K1280 acetylations within the hexon channel and the buttress domain of the capsid protein pUL86. In contrast, the K24 and K85 acetylations present in the N-lasso and the Johnson fold, regions known to function in pUL86 monomer interactions[47], were more prominent early in infection, suggesting a temporal regulation of these functional domains.

The presence of several acetylations early in infection prompted us to ask whether viral protein acetylation is already present in virions. Following virion enrichment using a sorbitol cushion and immunoaffinity isolation of acetylated peptides (Fig. 2a), we identified 105 acetylation sites on 37 HCMV proteins (Fig. 7c, Supplementary Data set 5). This prominent regulation of viral proteins by acetylation is perhaps striking when considering that HCMV has a low percentage of lysine residues encoded by its genome (Fig. 7b, Supplementary Data set 6). Although mammalian genomes encode for ~ 5–7% lysine residues, several herpesviruses, such as HCMV, have 2–3% lysine residues encoded by their genomes. This low lysine content is not a common feature observed in all viruses, as RNA and DNA viruses can have a range of lysine composition. Our results indicate that at least 5% of the lysine residues encoded by the HCMV genome are acetylated. Although acetylation sites are likely to still be uncovered for viruses, human and other species, so far the most in-depth analysis of acetylation in human cells has shown that ~ 2% of lysine residues encoded in the human genome are acetylated[12]. Therefore, the degree of HCMV protein acetylation seems to match and even surpass that known so

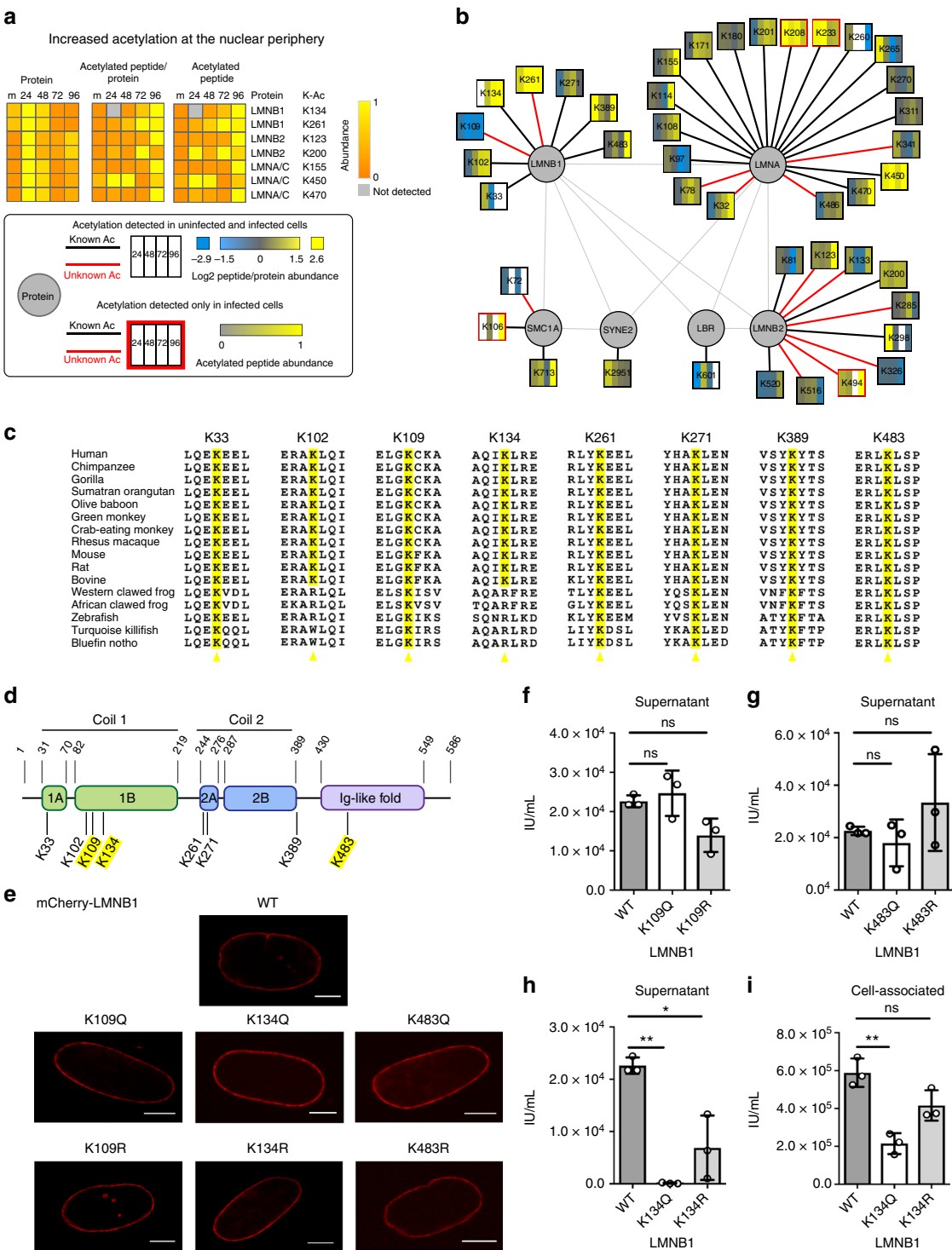

**Fig. 5** Functional acetylation at the nuclear periphery during HCMV infection. **a** Nuclear periphery proteins with increased levels of acetylation during infection, scaled from 0 to 1 (left, protein abundance; middle, peptide acetylation level normalized to protein abundance; right, peptide acetylation level) **b** Acetylation sites detected on lamin and lamin-associated proteins. **c** Sequence alignment of identified LMNB1 acetyl-lysines from selected animal species. The identified acetyl-lysine residues are highlighted and the six flanking residues were excerpted for representation. **d** Schematic of LMNB1 protein domains with detected acetylated sites; upper amino acids demarcate domains; lower, acetylated lysines; yellow, acetylation sites that were pursued in this study. **e** Representative images of MRC5 cells transfected with all seven mCherry-LMNB1 constructs indicated. All constructs localize to the nuclear periphery and do not alter nuclear morphology. Scale bar = 5 μm. **f–h** Effect of LMNB1 mutants on extracellular virus produced. Cells expressing either wild-type (WT) or mutant LMNB1 were infected, and the supernatant collected at 120 hpi was then used to infect a reporter plate of fibroblasts. Titer (IU/mL) was measured by staining the reporter plate for viral immediate early protein IE1. Average of three biological replicates ± SD using a two-sided Student's *t* test, *$p$ value ≤ 0.05, **$p$ value ≤ 0.01. **i** Cell-associated virus was collected and used to infect a reporter plate. Titer (IU/mL) measured via IE1 staining, **$p$ value ≤ 0.01. Source data are provided as a Source Data file

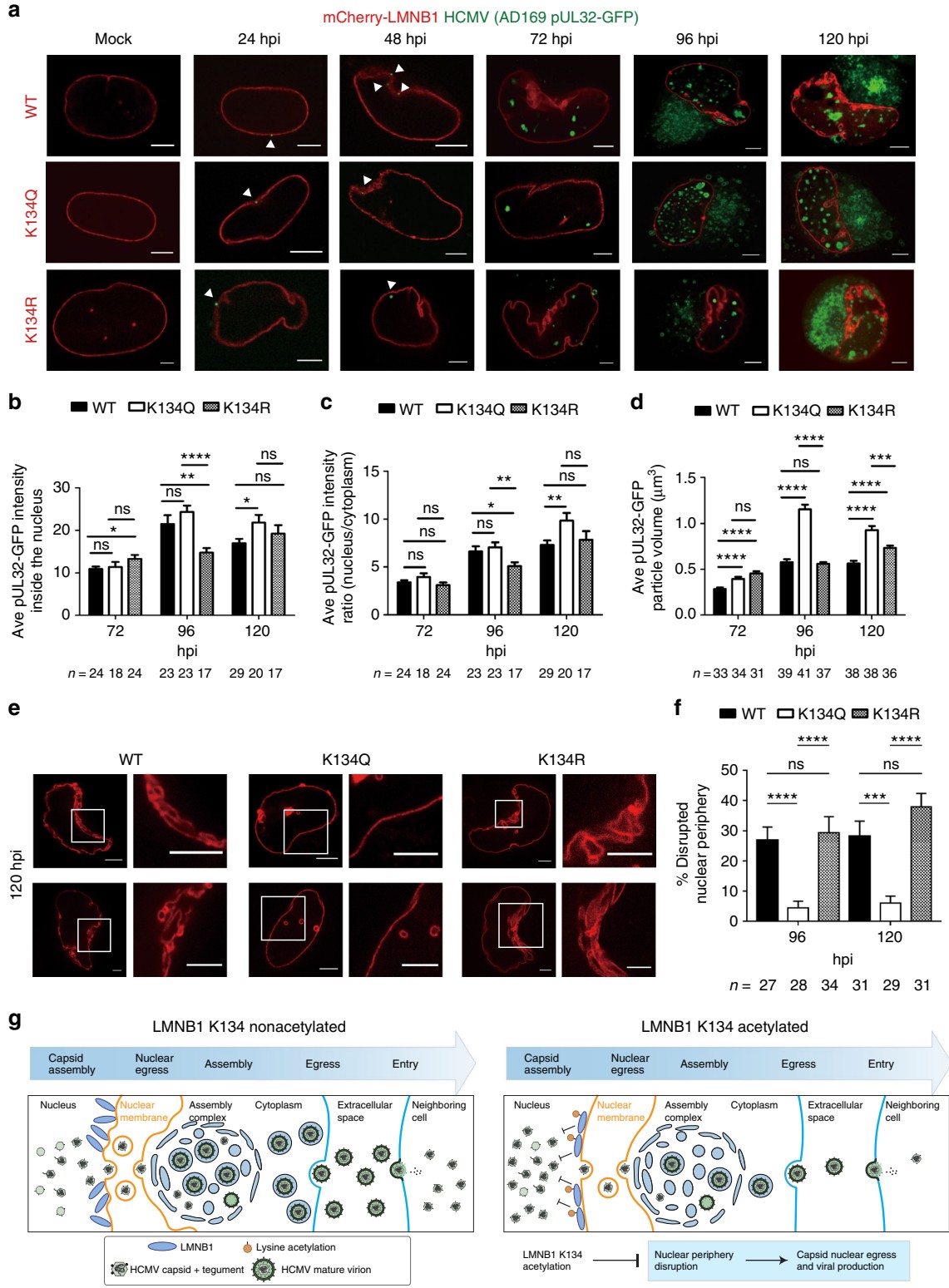

far for human proteins. Furthermore, the viral proteins detected as acetylated reside within different virion compartments (Fig. 7c), and numerous acetylations localize in known functional domains (Fig. 7d, Supplementary Fig. 11c). This suggests that acetylation may contribute to a range of virus–virus and virus–host interactions.

One viral acetylation identified both early in infection and in enriched virions was on the tegument protein pUL26. This augmenting viral protein is expressed with delayed early kinetics

during HCMV infection and acts both in transcriptional activation of the viral major immediate early promoter[46] and in enhancing virion stability[48]. pUL26 moves first to the nucleus and then to the AC as infection progresses[48]. The K203 acetylation resides in the C-terminal domain that is required for pUL26 functions (Fig. 7e)[46,49]. To investigate its function, we generated two HCMV mutant strains with point mutations for acetyl-mimic (glutamine) or charge-mimic (arginine) at K203. We tested both

**Fig. 6** LMNB1 acetylation impedes lamina disruption and viral capsid nuclear egress. **a** Live cell confocal fluorescence microscopy of infection time course in cells transfected with WT, K134Q, or K134R mCherry-LMNB1 (red) and infected with HCMV AD169 UL32-GFP (green). One slice through the center of the nucleus is shown. Representative images are shown; scale bar = 5 μm. **b, c** pUL32-GFP nuclear intensity in WT, K134Q, or K134R at each time point; the number of cells (*n*) is indicated. A two-sided Student's *t* test was used. **b** Mean pUL32-GFP intensity across the center three slices inside the nucleus + SEM. **c** The ratio of mean pUL32-GFP intensity inside the nucleus to outside the nucleus across the center three slices + SEM. **d** The volume of the pUL32-GFP particles was analyzed in a 3D rending of each Z-stack, and the number of analyzed cells (*n*) is indicated. The mean pUL32-GFP particle volume + SEM is shown. A two-sided Student's *t* test was used. *p value ≤ 0.05, **p value ≤ 0.01, ***p value ≤ 0.001, ****p value ≤ 0.0001. **e** Live cell confocal fluorescence microscopy of nuclei at 120 hpi for cells transfected with WT, K134Q, or K134R mCherry-LMNB1 (red) constructs. Left: one representative slice through the center of the nucleus. Right: zoom-in of the square on the left; scale bar = 5 μm. **f** Percentage of nuclear periphery disruption. Disruption in nuclear periphery are seen as infoldings, as visualized by mCherry-LMNB1 fluorescence across center slices. Mean percentage disruption + SEM is shown and cell numbers are indicated. A two-sided Student's *t* test was used; ***p value ≤ 0.001, ****p value ≤ 0.0001. **g** Proposed model for the function of LMNB1 K134 acetylation. Acetylation may stabilize LMNB1, which may inhibit nuclear periphery deformation and thereby impede HCMV capsid nuclear egress and infectious virus production. Source data are provided as a Source Data file

high multiplicity of infection (MOI = 3), at which all cells are simultaneously infected, and low MOIs (MOI = 0.25 and 0.05) that offer improved monitoring of defects in virus fitness during its spread. We found that at both high and low MOIs the acetyl-mimic (K203Q) of pUL26 resulted in reduced virus production (Fig. 7f–h). In contrast, the non-acetylatable charge-mimic (K203R) produced a higher titer than WT at the low MOIs tested, and this difference was significant at the lower multiplicity of infection (MOI = 0.05). These data suggest that acetylation inhibits the functions of pUL26 in supporting viral replication.

## Discussion

Protein acetylation has become well-recognized as a prevalent PTM that regulates core cellular pathways, including transcription, metabolism, and immune response. However, how human viral infections perturb the cellular acetylation landscape remains largely unexplored. No study has attempted to define global regulation of acetylation in either time or space at different stages of a viral infection. Using an integrative proteomics, mutagenesis, microscopy, and functional virology approach, we demonstrate that finely tuned temporal acetylation during HCMV infection affects processes integral to viral replication. As a common theme and perhaps surprising for an infection, we find that most changes in protein acetylation are not driven by alterations in protein abundances, thereby pointing to a specific modification-mediated regulation. Overlay of these temporal acetylome and proteome data sets with knowledge of dynamic protein sub-cellular localization during infection allows us to start to build a view of spatial changes in acetylations throughout HCMV infection.

One discovery from this assessment of organelle-specific acetylation is temporal regulatory events at the nuclear periphery. Among human viruses, HCMV is known to trigger dramatic changes at the nuclear periphery, with the formation of a kidney-shape nucleus and lamina disruption. The site-specific increase in LMNB1 acetylation late in infection and our functional studies led us to propose a model in which K134 acetylation acts in host defense by inhibiting the disruption of the nuclear lamina late in infection and thereby suppressing viral capsid egress from the nucleus (Fig. 6g). Although acetylation of nuclear lamina proteins was not previously investigated during infection, PTMs of intermediate filaments are known to modulate their function and stability[50]. Phosphorylation at Ser22 on LMNA/C disrupts this protein, facilitating primary envelopment of HCMV capsids at nuclear membrane infoldings that have been depleted of LMNA/C[43]. Although its regulation is not yet understood during HCMV infection, LMNB1 was previously observed at nuclear membrane infoldings[51]. The LMNA/C Ser22 is also conserved in LMNB1 (Supplementary Fig. 8), but it remains to be determined whether

this site can also be phosphorylated. Outside the context of infection, phosphorylation of LMNA/C was shown to provide a regulatory mechanism by triggering localized lamin disruption during mitosis[52]. It is possible that lamin acetylation, such as LMNB1 K134 acetylation, provides an additional mechanism for the local regulation of lamins. Notably, the same lysine residue was reported to be ubiquitinated, and LMNB1 is known be targeted for degradation upon ubiquitination. However, it is not yet known whether this specific site initiates its degradation, and this modification was not studied during infection[53,54]. Our K134R mutant studies suggest that an ubiquitination event at this site is not needed for the infection-induced local disruption of LMNB1. However, our results demonstrate that this residue plays an important role in lamin flexibility, with acetylation restricting the virus-induced disruption of the nuclear periphery. Acetylation may stabilize LMNB1 in a manner similar to the acetylation in the rod domain of another intermediate filament, keratin 8, which leads to the formation of dense perinuclear K8 networks[55]. As virus capsid nuclear egress requires the local disruption of the lamina, it is likely that acetylation, ubiquitination, and phosphorylation are finely tuned at the interface between host defense and virus replication. Since the virus-induced disruption is only local, and not global for the entire lamin, these modifications are not expected to have high stoichiometry. Of note, the majority of the previously detected phosphorylations on LMNA/C are present on the head, tails and inter-domain regions[56], whereas LMNB1 and LMNA/C acetylations localize to the coils of the lamin rod-domains and the Ig-like fold region. As PTMs can regulate numerous protein properties, including protein folding, stability, activity, interactions, and signaling, it is likely that these acetylations provide diverse regulatory mechanisms of lamin functions. For example, we find that the K134 site, which seems to have been acquired in mammalian species (Fig. 5c), is relevant for viral infection (Fig. 5h). In contrast, mutations of K109 and K483 do not impact virus production. Given the high evolutionarily conservation of these lysine residues across vertebrates (Fig. 5c), these sites may be involved in the general structural regulation of lamin. Alternatively, modifications at these sites could contribute to general host response mechanisms to nuclear remodeling through interactions with other factors and communication with cellular pathways. Altogether, our finding points to an additional layer of regulation of the nuclear periphery via acetylation, revealing the function of K134 acetylation in restricting virus replication. In addition to the effect on capsid nuclear egress, the fact that LMNB1 acetylation inhibits nuclear periphery disruption may impact multiple aspects of the late stages of HCMV infection. For example, as the curvature of the nucleus was shown to be necessary for the proper formation of the AC[37], the less-pronounced kidney shape in acetyl-mimic cells suggests that the stability of the resulting AC could be impaired. Consequently, the assembled viral particles may be partly defective and less

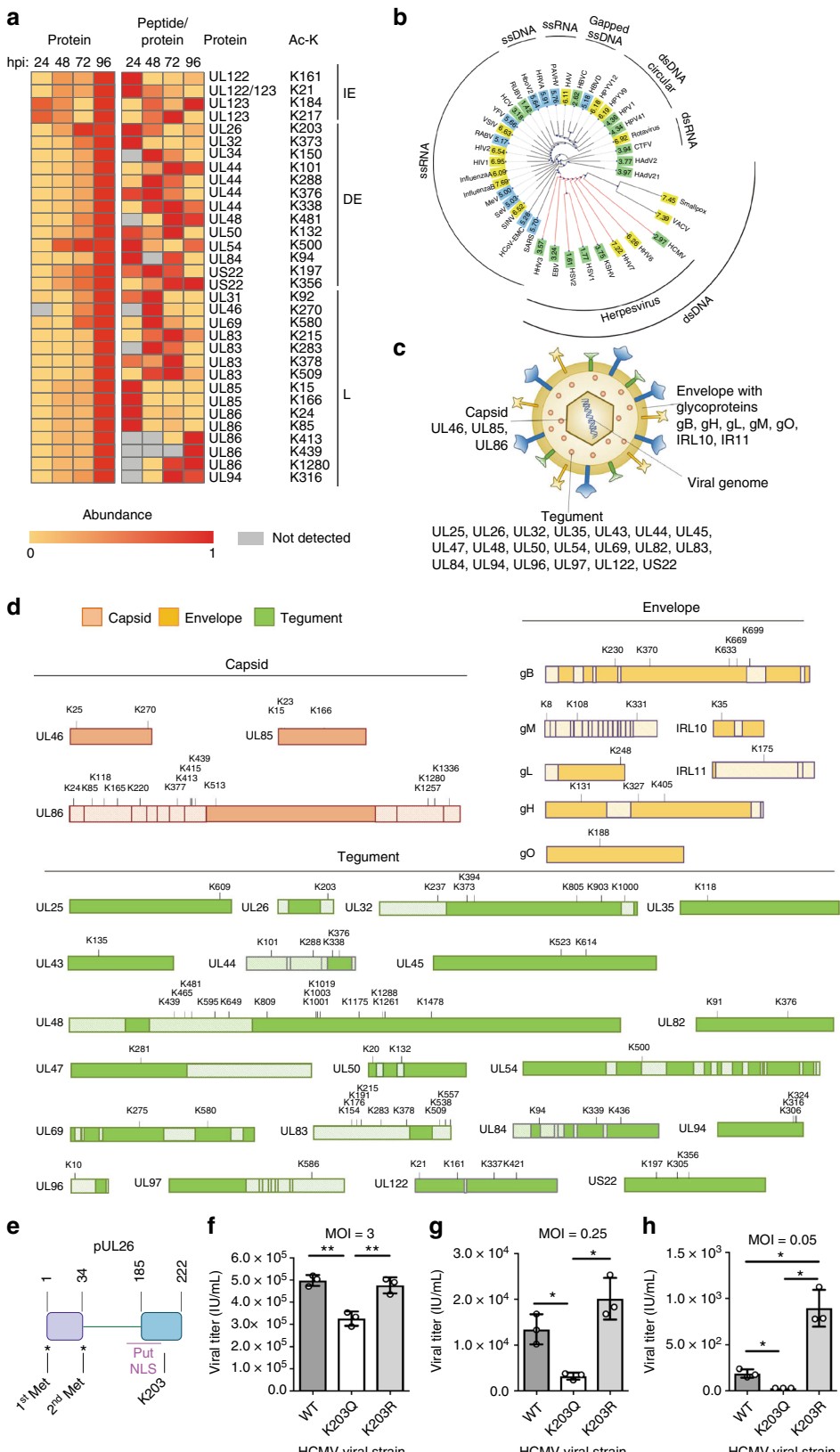

infectious. Therefore, the overall impact of LMNB1 K134 acetylation on virus titer could represent a cumulative effect on capsid nuclear egress and virion assembly.

Another identified spatial acetylation change was the upregulation of mitochondrial protein acetylation. This finding has

direct relevance to HCMV infection, as this virus requires modulation of cellular metabolism for fatty-acid synthesis and energy production[25]. For instance, increased K321 acetylation was observed on the pyruvate dehydrogenase subunit PDHA1, a critical component of metabolism that converts pyruvate to acetyl-

**Fig. 7** Prevalent acetylation on HCMV proteins, including pUL26 acetylation that restricts viral replication. **a** Abundances (scaled 0–1) of acetylated viral peptides and their corresponding proteins detected during infection (left, protein abundance; right, peptide acetylation levels normalized to protein abundance). Acetylated lysines and temporal stages of infection (IE (immediate early), DE (delayed early), and L (late)) are indicated. **b** Lysine content in the proteomes of different viruses. Viral families were clustered by proteome similarities (of representative strains). The lysine percentages of their proteomes were color coded with green (< 5%), blue (5–6%), and yellow (> 6%). **c** Identified acetylated viral proteins and their corresponding compartments in the virion. **d** Site-specific acetylations on capsid, envelope, and tegument virion proteins. Functional domains are indicated, and listed in Supplementary Fig. 5. **e** Schematic of pUL26 domains and putative nuclear localization signal (NLS). **f, g, h** Viral titers (IU/mL) determined by IE1 staining for infections at MOI 3 (f), MOI 0.25 (g), or MOI 0.05 (h) for WT and K203 mutant viruses (K203Q, K203R). Average of three biological replicates ± SD. A two-sided Student's $t$ test was used. *$p$ value $\leq 0.05$, **$p$ value $\leq 0.01$. Source data are provided as a Source Data file

coA. Acetylation at this site is known to decrease PDHA1 activity in uninfected cells[26]. As HCMV relies on increased flux through glycolysis, this PDH acetylation may serve an antiviral purpose. Other changes in mitochondrial acetylation may be proviral. A four-fold increase was found in the previously unknown K519 acetylation of ATP5B, a subunit of the ATP synthetase complex. Although ATP5B phosphorylation contributes to mitochondrial dysfunction in insulin resistance[57], there is no documented role for its acetylation. However, ATP5B is needed for successful budding of influenza A, the cell-to-cell transfer of HIV, and the efficient replication of arenaviruses and HSV-1[58–61]. Given the role of ATP5B in the replication of diverse viruses, the increased K519 acetylation may be proviral during HCMV infection.

The finely tuned site-specific regulation of protein acetylation during infection also highlights likely cross-talks between different PTMs. One evident interplay is the switch between acetylation and ubiquitination at the same residue. A canonical example is the p53 acetylation that blocks its ubiquitination and proteasome-mediated degradation, thereby supporting immune response[62]. Similarly, NF-κB acetylation by p300/CBP impedes its degradation, inducing its transcriptional-activating function[63]. As HCMV induces the degradation of numerous host defense factors to evade immune response[64], the acetylation-ubiquitination switch is likely critical during infection. The prominence of this is supported in uninfected cells, where nearly a third of human acetylation sites were reported to also be ubiquitinated[53]. Mitochondrial proteins are among these dually modified proteins, in agreement with our finding of infection-enhanced acetylations on TCA cycle proteins, including PDHB, DLAT, and DLD, at sites known as ubiquitinated. However, acetylation-ubiquitination interplays may also have proviral functions such as the acetylation of integrin beta-1 (ITGB1) K794, a cellular receptor for HCMV. Although the role of acetylation is unknown, ubiquitination on dimerized ITGB1 leads to its degradation[65], and K794 ubiquitination has been reported. Acetylation could inhibit ITGB1 ubiquitin-mediated degradation and support HCMV infection. Similar cross-talks are likely present between acetylation and other lysine PTMs. For example, the immune response factor RIG-I is known to be acetylated, ubiquitinated, ISGylated, and SUMOylated, and these PTMs drive different functions[66].

The presence of PTMs on viral proteins adds another layer of complexity to the role of acetylation during infection. Although recent work in insect and fish species revealed viral acetylomes as critical contributors to infection[67,68], acetylation of viral proteins during human pathogenic infections remains understudied. Most of our understanding of viral protein acetylation comes from work on individual proteins, such as influenza A NP, HIV Tat, and papillomavirus E2[6,8,10]. Here, we uncover numerous acetylations on HCMV proteins, none of which were previously identified. This prominent acetylation is perhaps surprising given the relatively low lysine content of the HCMV proteome when compared to several other viruses and mammalian species. We observe the presence of many acetylations within all virion compartments in enriched infectious particles.

HCMV tegument proteins are critical for promoting viral DNA replication and inhibiting host immune response. We identified 20 acetylated tegument proteins, and demonstrated the impact of pUL26 acetylation on virus replication. Among its multiple roles, pUL26 acts early in infection as a transcriptional activator of the viral immediate early promoter[48,49], represses non-canonical NF-κB activation[69] and the ISGylation of other viral proteins[70], and facilitates virion stability at the viral AC late in infection[48]. We find pUL26 K203 acetylation both in enriched virions and early in infection, and this site is within the domain critical for these pUL26 functions. Our observations that the acetyl-mimic mutant is defective in virus replication and that a non-acetylatable charge-mimic has higher fitness at low MOI suggest that acetylation impedes pUL26 functions, possibly as a host antiviral response. Another acetylated tegument protein is pUL83, a viral protein known to suppress interferon signaling by inhibiting the oligomerization of the DNA sensor IFI16[45]. Six of the identified acetylated sites are located within the pUL83 region that binds the pyrin domain of IFI16, and three other sites localize in the pUL83 C-terminal region that dissipates IFI16 oligomerization[45]. Whether these modifications contribute to either the pUL83 immune evasion function or a host inhibition of pUL83 remains to be determined.

In addition to numerous tegument acetylations, we found all HCMV capsid proteins to be acetylated, including the major capsid protein pUL86. Among its 13 acetylation sites, K118 in the Johnson fold domain[47] is conserved in all herpesvirus species. Although not previously discovered for HCMV, acetylations on capsid proteins have been reported for other viruses, including HSV-1, Adv2, polyomavirus, and the plant polerovirus[71–74]. Whether this common feature contributes to capsid stability remains to be seen. Our study also highlights the potential for acetylation-mediated regulation of HCMV glycoproteins. The gH K131 acetylation lies on the virion surface, thereby possibly influencing the glycoprotein interaction with a cellular receptor. Acetylations on gM fall within regions that interact with gN (K8, K108) and FIP4 (K331). Although not yet documented for herpesviruses, acylation of the influenza A HA protein is essential for virus replication[75]. It is possible that these extra-virion and intra-virion acetylations also facilitate HCMV entry into cells. In addition, we identified acetylations on the viral polymerase processivity factor pUL44. This protein forms a homodimer, making a C-shape clamp around DNA and binding to the viral polymerase, pUL54[76]. K101 and K280 acetylations fall within the region needed for pUL44 function, and K101 is in the short region essential for homodimerization[76]. As K101 acetylation increases with infection, it is possible that, similar to the known role of increased pUL44 SUMOylation in viral replication[77], this acetylation contributes to the stability of homodimers. We anticipate that our study, using the well-characterized HCMV laboratory strain AD169[78], will facilitate comparisons with previous work. In addition, this study should provide a foundation for future investigations with clinical HCMV strains[79] that contain a more complete viral genome and would allow further elucidation of nuanced protein acetylation in distinct tissues and

cell types during active replication or latency. As HCMV has a broad tropism[80], future acetylation studies in different cellular systems will enhance our understanding of HCMV biology and cellular pathology.

In summary, our study uncovers protein lysine acetylation as a highly regulated molecular toggle of protein function in both host antiviral defense and virus replication. We discover temporal and spatial alterations in the cellular acetylation landscape during the progression of infection, and show that HCMV proteins themselves can be regulated by acetylation. We provide biological insights into specific functions of cellular and viral protein acetylation, as well as a valuable resource of dynamic acetylation events. As the acetylated pathways that we identify are known to be hijacked by diverse human viruses during their replication cycles, we expect this work to serve as a stepping stone for understanding the broader role of acetylation in different types of viral infections and to aid in the identification of new targets for antiviral therapies.

## Methods

**Cell culture conditions**. Wild-type MRC5 human lung fibroblasts (ATCC) were cultured in Dulbecco's modified Eagle's medium (Life Technologies) supplemented with 10% (v/v) fetal bovine serum) and 1% (v/v) penicillin–streptomycin solution (GIBCO) at 37 °C in 5% $CO_2$. Cells were tested for mycoplasma contamination by ATCC by agar culture, PCR-assay, and Hoechst DNA staining. Cells were used within 10 passages. No further cell line authentication was conducted.

**Viral growth and infection**. HCMV strain AD169 was used as the wild-type strain for this study. HCMV strain AD169 pUL32-GFP was used for microscopy to mark viral particles. For infection, virus stock was diluted in culture medium and incubated on top of cells at the indicated MOI at 37 °C for 1 h. The media was then replaced with culture medium, and infected cells were incubated at 37 °C for the indicated lengths of time.

**Enrichment of acetylated peptides and sample preparation**. At each time point of infection, six 15-cm dishes of MRC5 cells were washed twice with ice-cold phosphate-buffered saline (PBS), scraped in 1 mL PBS, and combined into one microcentrifuge tube. Cells were spun down at $250 \times g$, washed with 3 mL PBS, and flash frozen in liquid nitrogen. Cell pellets were resuspended in preheated lysis buffer (50 mM Tris-HCl pH 8, 100 mM NaCl, 0.5 mM EDTA, 4% sodium dodecyl sulphate) and heated at 100 °C for 3 min. Cell suspension was cup horn sonicated for 30 pulses and heated until all clumps were dispersed (~ 6×). A BCA assay (Pierce) was conducted to ascertain protein concentration. Cell lysates were reduced and alkylated with 25 mM TCEP and 50 mM chloroacetamide at 95 °C for 5 min, and then subject to methanol/chloroform precipitation. The resulting protein pellets were air dried and stored at − 80 °C. Protein pellets were resuspended at 0.5 μg/μL by cup horn sonication in 25 mM HEPES pH 8.2. Proteins were digested with two additions of 1:200 trypsin:protein (Promega, Madison) and incubated at 37 °C with gentle rocking for 6 h and 12 h, respectively. The digested samples were acidified to 1% trifluoroacetic acid (TFA), incubated on ice for 15 min, and then spun down at $4000 \times g$ for 10 min at 4 °C. The supernatants were subjected to Oasis Column cleanup (Waters) per manufacturer's instructions. The samples were flash frozen in liquid nitrogen and lyophilized for 2 days. The lyophilized peptides were subjected to anti-acetyl lysine immunoaffinity purification using PTMscan kit (Cell Signaling Technology) per manufacturer's instructions. In brief, peptides were resuspended in 1 × immunoaffinity purification buffer. An aliquot of 50 μg of peptides was saved for whole proteome analysis, whereas the remainder of the sample was used for immunoaffinity purification per manufacturer's instructions. The enriched peptides were eluted by two rounds of 50 μL 0.15% TFA at RT for 10 min. Samples were adjusted to 1% TFA and subjected to SDB-RPS StageTip Desalting (3 M Analytical Bio-technologies) cleanup and were resuspended in 1% formic acid, 1% acetonitrile.

**Isolation of acetylated peptides from enriched virions**. Three roller bottles of confluent MRC5 cells were infected with AD169 HCMV P0 virus and infection was allowed to progress until the cells exhibited 100% cytopathic effect (CPE). Supernatant and cell-associated virus were collected. Cells were scraped into the media and the sample was centrifuged at 4000 rpm for 10 min. Supernatant was collected (supernatant virus), and cells were sonicated 5 × for 10 s in a cup horn sonicator to release cell-associated virus. This sample was centrifuged at 4000 rpm for 10 min to remove cell debris. The supernatant (cell-associated virus) was collected. Cell-associated virus and viral supernatant were combined and subjected to ultracentrifugation at $20,000 \times g$ for 1.5 h over a 20% sorbitol cushion to concentrate and enrich the virus. Viral pellets were resuspended in 1 × PBS, spun down, flash frozen, and stored at − 80 °C until ready to be lysed. Viral pellet lysis and acetyl enrichment were conducted in the same manner as for cells (see above).

**MS analysis**. Tryptic peptides were analyzed by nLC-MS/MS using an Ultimate 3000 nRSLC system (Dionex) coupled on line to a Q Exactive HF mass spectrometer (ThermoFisher Scientific) equipped with an EASYSpray ion source (ThermoFisher Scientific). Peptides were separated by reverse phase chromatography on an EASYSpray C18 column (2 μm × 75 μm × 50 cm) using solvents A (0.1% formic acid in water) and B (0.1% formic acid in 97% acetonitrile) at a flow rate of 250 nL/min with a continuous gradient from 3 to 35% B over 150 min. Acetyl-peptides from enriched virions were separated by a discontinuous gradient of solvent A and B at a flow rate of 250 nL/min as follows: 4% B to 14% B over 60 min, then to 25% B in 30 min, then to 50% B in 4 min over 90 min, on an EASYSpray C18 column (2 μm × 75 μm × 25 cm). Tryptic peptides from the enriched virion proteome were also separated by a discontinuous gradient of solvent A and B at a flow rate of 250 nL/min with the following modifications: 4% B to 14% B over 100 min, then to 25% B in 50 min, then to 50% B in 4 min over 150 min on the same column type. For each acetyl-lysine immunoprecipitation (IP), an MS1 survey scan was performed from 350 to 1800 m/z at 120,000 resolution with an automatic gain control (AGC) setting of 3e6 and a maximum inject time (MIT) of 30 ms. The top 10 precursors were selected for fragmentation and MS2 scans were acquired at a resolution of 30,000 with an AGC setting of 1e5, a MIT of 150 ms, an isolation window of 1.6 m/z, a fixed first mass of 100 m/z, a minimum intensity threshold of 1e5, peptide matching set to preferred, a loop count of 10, dynamic exclusion of 45.0 s, acquired in centroid. The whole proteome and whole virion MS1 acquisition settings were identical to those of the IP experiments. The MS2 acquisition settings for these samples differed from the MS2 for the IPs in the following ways: resolution of 15,000, MIT of 25 ms, loop count of 20, and isolation window of 1.2 m/z.

For processing of raw instrument files, MS/MS spectra were analyzed by Proteome Discoverer (PD v2.2, ThermoFisher Scientific). Mass accuracy was recalibrated offline using the spectrum files RC node and the Minora feature detection node was used for quantifying isotope features. The SEQUEST HT node was used to obtain peptide spectrum matches by searching fragmentation spectra against either a human and herpesvirus protein sequence database (UniProt-SwissProt, downloaded 2016-04) appended with common contaminants (22,241 total entries), or the same database containing reversed protein sequences, the latter of which was used by Percolator to filter peptide-to-spectrum matches (PSMs) to 1% false discovery rate (FDR) within each replicate. SEQUEST settings contained the following criteria: full enzyme specificity, maximum two missed cleavages, precursor mass tolerance 4 ppm, fragment mass tolerance 0.02 Da, static modification of carbamidomethyl cysteine (+57 Da), and dynamic modifications of methionine oxidation (+16 Da), deamidation of asparagine (+0.984 Da), protein N-terminal Met-loss + acetyl (− 89.03 Da), and acetyl-lysine (+42 Da). A PD consensus workflow was used to assemble individual samples and biological replicates into a unified result, which controlled PSMs, peptide, and protein FDR identification rates to ≤ 1% and label-free peptide and protein quantification (LFQ), detailed below. The LFQ analysis used the Feature Mapper node to sequentially perform retention time alignment (maximum RT window: 15 min, mass tolerance: 6 ppm) and mapping of peptide features between runs. The maximum RT and mass tolerance for feature mapping was automatically determined by the node, set as 2.0 min and 3.7 ppm, respectively, for the acetyl-IP data set, and 1.23 min and 2.8 ppm for the whole-proteome data set. Consensus matched features and associated intensity-based abundance measurements were linked to peptide identification at ≤ 1% FDR. For acetyl-peptide data, the un-normalized abundance measurements and acetyl-lysine site localization scores were exported to Perseus for downstream informatics analysis. For whole-proteome analyses, within the PD consensus workflow peptide abundances were normalized across all samples by the Total Peptide Amount method, then replicate sample groups with only one replicate value were removed from further quantification, and then imputation was performed using Replicate Based Resampling, which imputes missing values within each sample group using random values sampled around the median abundance value of replicates. Peptide abundance ratios (infected time point/mock) were calculated based on the Pairwise Ratio method, using the geometric median of all combinations of ratios within the respective replicates. Peptides are summarized into protein groups by strict parsimony rules, retaining proteins identified by two unique peptides, and protein ratios calculated from the median of peptide ratios, excluding peptides with modifications.

The ptmRS node in Proteome Discoverer was used to calculate the probability that the acetyl modification is present on each potential lysine within the peptide. Based on the observed fragment ions, each potential site is assigned a localization probability, which is translated into a percent confidence score. A minimum score threshold of 75% confidence was required, which removed modified peptides where the acetyl modification could not be confidently assigned. Raw abundances of the acetylated peptides from the whole-cell lysate acetyl-IP were imported into Perseus for further filtering by the following criteria: a peptide must have a 1% FDR and have been detected in at least two out of three replicates with a CV < 100% in at least one time point. All peptides passing these criteria were normalized by subtracting the median from all time points to correct for systemic variation in the data set. CV values were calculated prior to imputation. For acetylated cellular peptides, missing values were imputed based on a normal distribution for abundance analysis. $Log_2$ acetylated peptide abundance was calculated in Excel by dividing the acetylated peptide abundance at a given time point to the acetylated peptide abundance in uninfected (mock) cells. $Log_2$ acetylated peptide abundance normalized to protein abundance was calculated by dividing the acetylated peptide

abundance (normalized to mock) by the protein abundance (normalized to mock) for each time point. Missing values were not imputed for acetylated viral peptides from the WCL IP. For the virion enrichment acetyl-peptides, no abundance quantification was performed, but presence of the modified peptides was manually validated. Known viral protein motifs and localization were assigned based on literature searches.

To compare the protein abundance and the number of acetylated sites per protein, two distinct approaches were implemented. As not all proteins identified as acetylated from our acetyl-IP analysis were also detected in our whole proteome analysis, we first graphed the number of acetylation sites versus the average protein abundance in parts per million (ppm) sourced from PaxDB. Proteins in this database are scaled in ppm relative to the whole proteome of the reference data set, which in this case was the data set: *Homo sapiens*—whole organism (integrated). As a second approach, we used iBAQ scoring that uses information from our whole proteome analysis. Normalized protein abundances (summed precursor intensities) generated by PD 2.2 were divided by the number of theoretically observable tryptic peptides (8–30 amino acids in length, generated by Perseus). iBAQ abundances across infection time points were averaged to represent the average protein abundances during infection.

**Bioinformatics and functional pathway analysis.** Acetylated peptides were clustered by log2 abundance normalized to mock. Clustering was conducted using Cluster 3.0 with k-means clustering organized by genes with k equal to seven clusters and 100 iterations performed. Data were exported through Java Treeview and heatmaps were generated using Morpheus from the Broad Institute. Protein abundance and acetylated peptide abundance normalized to protein abundance were ordered to match the order for acetylated peptides normalized to mock. For GO Biological Process analysis, gene names of the proteins corresponding to the acetylated peptides from each cluster were submitted to the Reactome Cystoscope plug-in (version 6.1.0). The Reactome GO BP analysis was used and terms were considered significant if they had a *p* value ≤ E-5. Uniprot accession numbers were submitted to STRING (version 10.5) to generate protein–protein functional networks. STRING networks were imported into Cytoscape (version 3.6.0) for overlay of quantitative data. Known acetyl lysine sites were annotated from the PhosphoSite Plus database.

**Localization analysis.** Each acetylated peptide was matched to its annotated localization by time point in the HCMV organelle proteome study recently published by our lab[20]. The accession numbers of proteins that were not found in[20] but were detected as being acetylated in our study were imported into Uniprot. The primary Uniprot localization was used to assign the localization of these proteins. All translocating protein information was obtained from the HCMV organelle proteome study. Clustering and cluster visualization were conducted as for the entire acetylated peptide data set with the only modification being adjustment of the number of k clusters based on the number of acetylated peptides in each organelle subset (cytoplasm k = 7, ER k = 5, Golgi k = 3, lysosome k = 3, mitochondria k = 7, nucleus k = 7, peroxisome k = 2, PM k = 5). Localization confidence scores for translocating proteins were imported from Jean Beltran, et al. 2016.

**Motif analysis of acetylated peptides.** Acetyl-peptide sequences were extended using PEPTIDEXTENDER (http://schwartzlab.uconn.edu/pepextend) with *H. sapiens* proteome to obtain peptides consisting of two sequences of six amino acids flanking the acetylated lysine (position 0). The extended peptides were analyzed by IceLogo. Percentage difference scoring method and a *p* value cutoff of 0.01 were applied, and *H. sapiens* proteome database was selected as the reference set. The IceLogos representing percentage differences at each position are plotted.

**Viral phylogenetic tree generation.** Viral proteomes were retrieved from Uniprot, and their lysine frequencies were calculated by Python. Alignment-free analysis of viral proteomes was conducted using alfpy package, and the generated distance matrix was subjected to neighbor method in phylip (Felsenstein, J. 1989. PHYLIP—Phylogeny Inference Package (Version 3.2). Cladistics 5: 164-166.) to create a tree file, based on which the phylogenetic tree was visualized by the ete3 package in python.

**mCherry-LMNB1 mutagenesis.** The mCherry-Lamin B1-10 construct (Plasmid #55069) was purchased from Addgene. Site-directed mutagenesis was conducted (10 × HF Buffer, 2.5 mM dNTPs, 10 μM Primer 1, 10 μM Primer 2, 25 ng/μL template plasmid, Phusion polymerase) via PCR followed by DpnI parental plasmid digest and electroporation into electrocompetent *Escherichia coli*. Positive mutants were selected by kanamycin resistance and confirmed by sequencing. Primers used for lysine mutagenesis to glutamine or arginine are as follows: LMNB1 K109Q (5′-CTGGGCCAGTGCAAGGCGGAACACG ACCAGCTGC-3′ and 5′-CTTGCACTGGCCCAGCTCGATCTGCAGCTTGGCG-3′), LMNB1 K109R (5′-CTGGGCAGGTGCAAGGCGGAACACGACCAGCTGC-3′ and 5′-CTTGCACCTGCCCAGCTCGATCTGCAGCTTGGCG-3′), LMNB1 K134Q (5′-CAGATCCAGCTTCGAGAATATGAAGCAGCACTCAATTCG-3′ and 5′-T CGAAGCTGGATCTGGGCGCCATTAAGATCAGATTC-3′), LMNB1 K134R (5′-CAGATCAGGCTTCGAGAATATGA AGCAGCACTCAATTCG-3′ and 5′-T CGAAGCCTGATCTGGGCGCCATTAAGATCAGAT TC-3′), LMNB1 K483Q

(5′-AGTTATCAATATACCTCAAGATATGTGCTGAAGGCAGGCCAGAC-3′ and 5′-GGTATA TTGATAACTGACTGATGTGTCTCCAATTTTTCTG ATCATCTCCCAG-3′), and LMNB1 K483R (5′-AGTTATAGATATACCT CAAGATATGTGCTGAAGGCAGGCCAGAC-3′ and 5′-GGT ATATCTA TAACTGACTGATGTGTCTCCAATTTTTCTGATCATC-3′).

**LMNB1 sequence alignment.** Protein amino-acid sequences were retrieved from Uniprot for 16 different species, and aligned by MultAlin and modified by ESPript3.

**Transient transfection and live cell imaging analysis.** In total, 300 ng of each mCherry-LMNB1 construct was incubated with 1 μL Xtremegene in OMEM at room temperature for 20 min. This mixture was added to MRC5 cells in OMEM and transfection was permitted to proceed for 5–6 h. OMEM was replaced with complete media. Uninfected cells were imaged the following day and immediately infected with an AD169 HCMV strain that expresses pUL32-GFP. Cells were live imaged in both red (mCherry-LMNB1) and green channels (pUL32-GFP) every 24 h for the course of infection. Imaging was performed on a Nikon TIRF with spinning disc using a × 100 oil-immersion objective. NIS-Elements software was used for acquisition and Z-stacks were collected. ImageJ was used for image analysis.

To assess the HCMV capsids inside the nucleus, the image Z-stacks taken at 72, 96, and 120 hpi were converted to separate slices after background subtraction, and the three center slices of each Z-stack were selected for quantification. In each slice, the region within the mCherry-LMNB1 boundary in the red channel defined the nuclear region, and was automatically selected in each slice by ImageJ script. The pUL32-GFP signal within this region was used to characterize the nuclear capsid parameters, and the pUL32-GFP signal outside this region was used to characterize the cytoplasmic capsid parameters. The mean pUL32-GFP intensity values of three slices from each stack were averaged by condition, and the mean nuclear/cytoplasmic ratio of pUL32-GFP was calculated by dividing the pUL32-GFP intensity within the nuclear region to that within the cytoplasmic region. To quantify the volume of pUL32-GFP particles, maximum projection was conducted for mCherry-LMNB1 in the red channel to define the nuclear region for each image, and 3D reconstruction of pUL32-GFP particles in the green channel was conducted within the nuclear region. The volumes of pUL32-GFP 3D objects were averaged for each condition. Steps involving Z-stack processing, selection of region of interest, intensity measurement, and 3D particle reconstruction were conducted by ImageJ scripts, and images were treated identically. For each of these analyses, mean + SEM comparisons were made. To assess nuclear periphery disruption in the presence of WT, K134Q, and K134R LMNB1 constructs during infection, the percentage of nuclear peripheral disruption was measured at 96 and 120 hpi based on the main center slices in each Z-stack. The average percentage disruption and SEM were calculated.

**AD169 BAC mutagenesis and virus production.** All UL26 mutants were derived from the WT AD169 BAC. The point mutant constructs are UL26 K203Q and UL26 K203R. Red recombineering was used to construct the viral mutants by PCR recombination as previously described. In brief, a PCR amplified cassette from the pEPkan-S vector with the appropriate UL26 mutagenic flanking sequences was electroporated into *E. coli* GS1783 cells containing the AD169 WT plasmid. Recombinant colonies were screened by growth on kanamycin, PCR amplification of the insert region, and agarose gel electrophoresis. Bacterial stocks with positive integrate sizes were grown in the presence of chloramphenicol and L-arabinose to resolve the co-integrates. Successful resolution of co-integrates was assessed by selection in the presence of chloramphenicol and L-arabinose. Positive colonies were replica plated onto chloramphenicol and chloramphenicol + kanamycin plates. Colonies that survived on chloramphenicol but died on chloramphenicol + kanamycin were picked for PCR amplification and agarose gel electrophoresis size assessment. All mutations were confirmed by sequencing. The primers for generating mutant viruses were as follows: UL26 K203Q (5′-TACGTGAGCC GCGTGCTGCTACGTCACCGTGTGAC-GCCGGGCCAGCAAGAAAT CACCGACGCGATGTAGGGATAACAGGGTAATCGATTT-3′ and 5′-TG GCACGTTGCCGGCTTCGAACATCGCGTCGGTGATTTCTTGCTGGCCCG G-CGTCACACGGTGACGGCCAGTGTTACAACCAATTAACC-3′), UL26 K203R (5′- TACG-TGAGCCGCGTGCTGCTACGTCACCGTGTGACGCCGGG CAGGCAAGAAATCACCGACGCGATGTAGGGATAACAGGGTAATC GATTT-3′ and 5′-TGGCACGTTGCCGGCTTCGA-ACATCGCGTCGGT GATTTCTTGCCTGCCCGGCGTCACACGGTGACGGCCAGTGTTA CAACCAATTAACC-3′). Mutagenized AD169 BAC constructs were electroporated into confluent MRC5 cells in a 10 cm dish. Cells were grown until they expressed 100% CPE. Supernatant and cells were harvested and stored as P0 stocks. Confluent roller bottles of MRC5 cells were inoculated with the P0 stocks and grown for ~ 2 weeks until cells expressed 100% CPE. Viral stocks were harvested as described above. Viral titer (PFU/mL) was assessed via TCID50.

**Viral titer assay by IE1 staining or TCID50.** MRC5 cells were infected with WT HCMV at the indicated MOIs and supernatants or cells were collected at 120 hpi and stored at − 80 °C. A reporter plate of WT fibroblasts was cultured until confluent. Collected supernatants were thawed and used directly. Cell-associated virus was extracted from the frozen cells by thawing and sonicating 5 × 10 s followed by 1 min

centrifugation at 5000 rpm to pellet cell debris. The resulting supernatant of cell-associated virus was then used for titering assay. Virus to be titered (either supernatant or cell-associated) was diluted in a serial manner and added to the reporter cells. After 24 h, the reporter cells were washed 3 × with cold PBS and then incubated with −20 °C methanol at −20 °C for 15 min. Cells were washed with PBS 3 × and allowed to rehydrate. Cells were subsequently blocked with 3% bovine serum albumin (BSA), incubated with IE1 primary antibody (as in Lorz, K. et al.[81], Ms, 1:40 in 0.3% BSA), washed 3 × with PBS with Tween, incubated with secondary antibody (Alexa 488 goat anti-mouse IgG (H + L) (Life Technologies, A-11001), 1:1000 in 0.3% BSA), incubated with 1:1000 4′,6-diamidino-2-phenylindole to mark all cellular nuclei, and washed 3 × with PBS with Tween. Operetta imaging system (Perkin Elmer) was utilized to visualize the reporter plate and count the IE1-positive cells. Thirteen fields of view per image were acquired, and the total number of IE1-positive cells across these fields was used to calculate infectious units/mL (IU/mL). Experiments were done in biological triplicate, and a two-tailed student's t test was used to assess significance. Viral titer for the LMNB1 WT and K134Q mutant was also assessed by $TCID_{50}$ in which MRC5 cells expressing LMNB1 constructs were infected with HCMV. Supernatants were collected at 120 hpi and serially diluted to infect a reporter plate of WT fibroblasts. The presence of plaques was assessed after 12 days.

**Statistics**. Statistical analyses were performed using a two-tailed student's t test unless otherwise stated. Data are presented as mean + SD or mean + SEM as indicated. Number of replicates is noted in the text and figure legends, and these numbers of replicates are in line with other literature reports. Data meet the requirements of the tests used.

## Data availability

The MS proteomics data have been deposited to the ProteomeXchange Consortium via the PRIDE partner repository with the data set identifier PXD009839. The source data underlying Figs. 5f–i, 6b-d, f, and 7f–h are provided as a Source Data file. Any additional data are available from the corresponding author upon request. A Reporting Summary for this article is available as a Supplementary Information file.

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

## Acknowledgements

We thank T.M. Greco and J.D. Federspiel for their advice on the anti-acetyl lysine IP. We thank K.C. Cook, J.D. Federspiel, and T.M. Greco for reading the manuscript. We are grateful for funding from the NIH (GM114141) and a Mallinckrodt Scholar Award to I. M.C., a National Science Foundation Graduate Research Fellowship (NSF-GRFP DGE-1656466) to L.A.M., and the National Institutes of Health NIGMS (T32GM007388).

## Author contributions

L.A.M., X.S. and I.M.C. designed research. L.A.M. and X.S. performed experiments. L.A. M., X.S. and I.M.C. analyzed data. L.A.M., X.S., and I.M.C. wrote the manuscript.

## Additional information

**Competing interests:** The authors declare no Competing Interests.

