## [Peer Review File · Nature Communications]

Reviewers' Comments:

Reviewer #1:

Remarks to the Author:

Lysine acetylation is a widespread and functional PTM in eukaryotes, however, its substrates and function in viral proteins remain largely unexplored. In this manuscript, the Cristea team performed a quantitative acetylome analysis to compare the acetylation profile on both cellular and viral proteins throughout the HCMV replication cycle in primary human fibroblasts, and successfully identified numerous acetylation sites were significantly changed under virus production. The text is well written and the authors present a valuable compilation of information about acetylation sites and give new insights into the role of protein acetylation in HCMV replication. I recommend that the manuscript be accepted pending the following revisions:

1. The number of samples used in each study is not clearly described. I was thinking there were three biological replicates for the proteomics study, but according to Supplementary Table 1, there seems to be only two replicates for their acetyl-lysine IP and MS analysis. I would suggest at least three biological replicates for quantitative proteomic studies. This is very important for confident identification of differentially expressed protein and their PTMs.

2. Please include description of how the reliability of the quantification measurements between biological/technical replicates was handled (statistical analyses, estimates of uncertainty, etc.)

3. I have tried my best to download the MS proteomics data using the accession number PXD009839 from the ProteomeXchange Consortium, however, no projects found for search term PXD009839. Without having access to their raw data or peak lists I cannot tell the quality of their MS experiments.

4. The K to Q mutations of LMNB1 mimic constitutively acetylated state while conserving the same charge, can the authors also create K to R mutants which will mimic non-acetylated state of LMNB1? This would nicely support the roles of the acetylation sites analyzed.

5. Can anything be said about the relative stoichiometry of the K-acetylation found? For example, what is the ratio of K109 acetylation in K134Q/K483Q mutants or the ratio of K134 acetylation in the other two mutants? I assume the relative level of modification is important to support the functional results for LMNB1, which may affect nuclear egress of viral capsids.

6. Is there a plausible explanation to describe why K109/483 do not have the same functional importance, yet these sites are still modified?

7. The specific criteria in the confidence determination for peptide modification need to be explained in this manuscript.

8. Page 7, line 135: Please include the detailed description of "ppm".

Reviewer #2:

Remarks to the Author:

Murray et al provide an interesting study and resource making the first systematic analysis of changes in acetylation of viral and host proteins throughout infection with human cytomegalovirus (HCMV). There is increasing evidence that lysine acetylation is an important regulatory PTM that can influence a multitude of host responses, and can have roles in innate immunity. Their study is also interesting since relatively little is known about acetylation of viral proteins in general, and very little is known about acetylation of HCMV proteins.

The authors describe an approach to measure changes in acetylation that sensibly includes

correction for overall changes in expression of proteins at a whole cell level. They further attempt to define acetylation over time on a subcellular basis, using results of their previous proteomic study that assigned proteins to subcellular structures based on density gradient fractionation (Beltran et al Cell Systems 2016). Finally, they find interesting new acetylation sites in nuclear lamins, and provide some functional data that relates these modifications to viral replication via inhibition of capsid egress. They also identify K203Q acetylation of HCMV UL26 as an important PTM that modulates viral replication. The study is in general well performed, and the manuscript well written. However, I have several concerns that should be addressed, under 'major points' and 'minor points' below:

Major points

1. The number of proteins quantified is relatively modest, due to the lack of peptide fractionation. In their whole cell lysate analysis, only 1065 proteins were quantified (Table S1). Had peptide or protein pre-fractionation been performed (for example by running lysates onto a gel and cutting into 6 fractions, as per their 2016 study), many more proteins would have been quantified. Similarly, the number of peptides in their acetyl analysis would have increased proportionally. For the latter, it may be that this would have presented practical difficulties due to the sheer amount of material required; however this should at the least be explicitly discussed, and a detailed table provided showing numbers of proteins and peptides quantified in each part of each experiment (i.e. each time point in duplicate for the whole cell lysate analysis, the overlap to generate the protein table S1, and the overlap between experiments – e.g. between whole cell lysate and acetyl enrichment).
2. The authors employ 'imputed values' to account for missing values, where a protein is not quantified in every analysis. These appear to make a minor contribution to the WCL analysis, however are more important in their acetyl study. I would suggest (a) providing a supplementary table showing the number of imputed values per experiment as a proportion of total, and (b) in Table S2, highlighting in columns T-AC which values are imputed (e.g. in colour). I realise this is shown in AD-AM by cells lacking data, however it is important for the use of this data by others to recognise which values were not actually measured.
3. The normalisation to absolute whole cell lysate protein abundance is important, however can potentially be confounded by proteins that were only measured by single peptides, or by the use of imputed values for protein-level measurements. This should also be discussed, and a column added to Table S2 to reflect the number of peptides measured per protein at a whole cell level. Where imputed whole cell values were used should also be shown.
4. The authors derive confidence in their whole-cell data from an analysis of up- or down-regulated host pathways. However, this could be performed in greater depth and precision by comparing their current temporal whole-cell data to their previous 2016 published whole cell data set. From the cluster plots, their dynamic range of changes even by 96 h of infection seems quite modest (from Table S1, only ~20 proteins up or down-regulated >4-fold) - was this seen before?
5. "To determine temporal changes in acetylation, we focused on identifications that were consistent between biological replicates at the different time points, and for which we could accurately determine levels" – please explain this statement, and justify in detail any data that has been discarded in their analyses.
6. "Furthermore, by monitoring in parallel the corresponding protein abundance levels (Fig. 2b, left) and normalized acetylation versus protein levels (middle), we found that a majority of the identified alterations in site-specific acetylations are independent of protein abundance changes." Comparing the left to right hand panels of Figure 2b, it seems that many of the upregulated acetylated peptides are also upregulated at the whole protein level, and a proportion of the

downregulated peptides are similarly downregulated. To justify their statement, some form of quantitation and statistics should be provided (a series of dot plots for each time point might help). Specific proteins (e.g. IFITs etc) could then be indicated on these plots, which would make this much easier to understand.

7. "Strikingly, nearly all of the sites on IFIT2 decline in abundance compared to their levels at 24 hpi. This points to a possible rapid acetylation-dependent IFIT2 response that is specific to the early stages of infection." I disagree – their time-course profiles of whole protein IFITs 1,2,3 also decline. It would be useful to include time-course plots showing (a) whole protein abundance (ALL peptides – already excluding modified peptides) and (b) acetylated peptide abundance. If their assertion is at all justified, there should be a statistically significant difference between both groups.

8. Their conclusions about the correlation between site-specific acetylations and organellar translocations seem at the least highly speculative. For example, MGST3 apparently translocates from the mitochondria to the peroxisome at 48 h of infection then back again. Furthermore although there is no change in acetylation upon the initial translocation, acetylation decreases for the peroxisome to mitochondrion shift. The authors show this and several other similar examples (which presumably represent their best data). They speculate "It remains to be seen whether acetylation drives the changes in localization or whether the localization induces a change in acetylation." In fact neither may be the case. Since changes in peptide acetylation within any given organelle hasn't actually been measured, it's impossible to know whether the acetylated form of the protein translocates, with the non-acetylated form of the protein staying put, or if the changes in acetylation are actually completely irrelevant to organellar location. The authors could potentially re-research the data from their 2016 publication with a variable acetyl modification. Although there was no acetyl-specific enrichment, one might expect to identify some acetylated peptides anyway. This might (or might not) help to support their conclusions. Alternatively, if any anti-acetyl antibodies are available for any of the proteins of interest, immunofluorescence could provide supportive information. Furthermore, could some measure of the reliability of their assessment of protein localisation / movement (from their previous publication) for these examples be provided in the supplementary (e.g. all the peptide profiles from each protein)? A protein translocating at a single time point between mitochondrion and peroxisome or Lysosome and ER/Golgi seems surprising and gives concern that the measurement of location at this time point may not be correct. Without any supportive data, the conclusions from this section, particularly "By integrating our acetylome dataset with spatial-temporal information of protein localization during HCMV infection, we provide a perspective of how site-specific, temporal acetylation can be regulated at a subcellular level to impact viral replication" are a very considerable overreach.

9. Although the findings with regards to the K132Q mutant are very interesting (200 fold drop in viral titre), the conclusions on mechanism (Figure 5b-c) are dubious. The only differences in UL32-GFP intensity in the nucleus they see are relatively modest changes at 120 h post infection – by which point infected fibroblasts if still alive would be falling apart due to the terminal stage of viral infection. Further data to better justify their mechanistic conclusions are warranted.

10. This study employs the laboratory adapted HCMV strain AD169. Although this is convenient to work with and can easily be grown to high titre, it is less similar to a wild-type strain than less extensively passaged HCMV isolates, particularly as it lacks ~20 genes in the UL/b' locus. The authors should at least acknowledge this as a potential caveat.

11. A number of the inferences in the text seem overstated – e.g. "This points to a likely major regulation of metabolic pathways during infection"; "at least 5% of the lysine residues encoded by the HCMV genome are acetylated, providing evidence that viral protein acetylation is likely a prominent regulatory event."

Minor comments:

1. For Table S1, please explain in the legend the meaning of 'high', 'peak found' and 'not found'.
2. Figure 3a – the cluster plots should be plotted on the same scale (at least for the acetylated peptide / protein vs acetylated peptide; I realise different scales may be required for each organelle).
3. Figure 3b – this is difficult to understand. Could this be more clearly explained in the legend?
4. Supplementary Fig 4b should read Supplementary Fig 3b line 305.
5. Line 362 should read Fig. 6c not Fig. 6b
6. Their conclusions in Figure 1c comparing to average protein abundances from PaxDb could be improved by estimating actual protein abundances / cellular concentrations using e.g. the Histone Ruler (see Wisniewski et al), iBAQ etc.
7. Figure 2a – please provide a supplementary table of all biological processes quantified in this analysis.
8. For the localisation analysis, please provide details of in which cases this was derived from their previous study, and which from Uniprot.
9. Please provide a username and password for review of the deposited RAW files.
10. Statistical analyses seem appropriate, apart from where extra analysis is needed as detailed above.

Reviewer #3:

Remarks to the Author:

In their current paper Murray et al use a mass-spectrometry screening approach to quantify the level of a common protein modification - lysine acetylation - during CMV infection of human fibroblasts (MRC5). Their approach is methodologically sound and their results are novel. While the manuscript leans heavily towards the descriptive side, this is not surprising given the scarcity of research on protein acetylation. The major novelty of the paper is in the global identification of acetylation events on both the human and viral proteins, which are likely to open the door for more mechanistic investigations. The authors further elaborate on two specific findings; the acetylation on the host protein Lamin B and the acetylation of the viral protein pUL26.

For Lamin B, the authors identify 3 lysine residues for which acetylation increases during the course of infection. They overexpress acetylation-mimetic mutants of these sites (replacing the lysine with glutamine) and find that overexpression of one mutant (K134Q) leads to a 200-fold reduction in the number of infected cells.

For pUL26, the authors identify a single lysine acetylation (K203). They construct two mutant viruses - an acetylation-mimetic mutant, K203Q, and a charge-mimetic mutant, K203R and find that the acetylation-mimetic mutant is less fit, showing fewer infected cells, while the charge-mimetic is as fit as wild type.

Major concerns:

1. Viral titrations: The authors use IE1 staining and report the results as infectious units/ml. As infectious units usually refers to the number of plaque producing particles this is misleading. This is especially important in cases where the effect might be later than viral entry and IE expression, such as in the case of Lamin B possible effect on nuclear egress. The preferable approach would be the use of plaque assays to calculate the results as PFU/ml. Alternatively, the authors can rewrite the relevant paragraphs to say that they measure the effect on IE expression.
2. The role of Lamin B acetylation in CMV infections:
 - a. The fact that the same residue, K134, is known to be ubiquitinated during CMV infection and that Lamin B degradation is also needed for CMV infection makes it very hard to deduce anything about the specific role of the acetylation in this process. The authors propose that the role of the K134 acetylation is to compete for ubiquitination, thus preventing K134 ubiquitination and Lamin B degradation, but they offer no experimental support for this model. I am unsure how the authors

can convincingly show a role for the K134 acetylation, since any mutation of K134 will also affect its potential ubiquitination. At the very least, the authors should test the role of the charge mimetic mutation (K134R) on CMV infection. If this mutant is also more resistant to CMV replication this would suggest that the K134Q mutant effect is likely due to its inability to be ubiquitinated, and not its acetylation-mimetic nature.

b. It is not clear how the experiment in Fig. 4d was done. Did the authors infect cells expressing the mutant Lamins and stained them for IE1 directly, or did they collect viruses from the cells (and if so was this only supernatant or also cell-associated?) and then titrate them using the IE1 staining? There is obviously a major difference between the two possibilities.

c. In Fig 5b-c the authors quantify the relative abundance of CMV in the nucleus and cytoplasm. They find a relatively small effect (<20%) of the K134Q mutant on this ratio, and only at 120 hrs post infection. This small effect does not seem to explain the major effect of this mutation on viral titer (Fig. 4d – 200-fold difference). The authors should acknowledge this discrepancy and offer some interpretation. As it stands, I would think that the effect of the mutation is most likely at a step later than nuclear egress.

3. The role of pUL26 acetylation: The authors generate two mutant viruses, one that carries the acetylation-mimetic mutant, K203Q, and one that carries the charge-mimetic mutant, K203R. At MOI of 3 and 0.25, the K203Q mutant is less fit than the wild-type, while the K203R is not statistically different than wt. The authors conclude that: "These data suggest that acetylation inhibits the functions of pUL26 in supporting viral replication" (Line 378). However, if the acetylation played an inhibitory role, one would expect that the K203R, which is unable to go acetylation, would have a higher fitness than the wt virus. There is in fact a small increase in the K203R fitness at the lower MOI (0.25), which is not statistically different. I would advise the authors to repeat these experiment in much lower MOI, such as 0.01, in which case smaller fitness differences are likely to be more visible. In any case the authors should address this point, as currently their conclusion is not fully supported by the data they present.

Minor concerns:

1. The manuscript requires some editing. For examples: the text references Fig S4 instead of S3 (Line 305), the supplementary files contains editing remarks.

We sincerely thank the editor and reviewers for the careful assessment of our manuscript and for their insightful comments and recommendations. Based on the reviewers' comments, we have performed a series of additional experiments that address all of the reviewers' concerns. Specifically, we have performed a third biological replicate for both the acetylome and proteome studies at all time points of infection, have generated new mutant constructs, and have added more microscopy and virus titer analyses. As a result, we have added 10 new figure panels in the main manuscript (Fig. 3c, Fig. 4c, 4e-i, Fig 5e-f, Fig. 6h), 10 new supplementary figures/panels (Figs. S1f, S2, S3, S4, S5b-d, S7b, S8, S9, S10, S11a-b), and revised all remaining figures and text to include the new data. All new data support our initial interpretations and conclusions. In what follows, we present a point-by-point answer to the reviewers' comments. Our responses are marked with "**>Response**".

Reviewers' comments:

Reviewer #1 (Remarks to the Author):

Lysine acetylation is a widespread and functional PTM in eukaryotes, however, its substrates and function in viral proteins remain largely unexplored. In this manuscript, the Cristea team performed a quantitative acetylome analysis to compare the acetylation profile on both cellular and viral proteins throughout the HCMV replication cycle in primary human fibroblasts, and successfully identified numerous acetylation sites were significantly changed under virus production. The text is well written and the authors present a valuable compilation of information about acetylation sites and give new insights into the role of protein acetylation in HCMV replication. I recommend that the manuscript be accepted pending the following revisions:

1. The number of samples used in each study is not clearly described. I was thinking there were three biological replicates for the proteomics study, but according to Supplementary Table 1, there seems to be only two replicates for their acetyl-lysine IP and MS analysis. I would suggest at least three biological replicates for quantitative proteomic studies. This is very important for confident identification of differentially expressed protein and their PTMs.

>Response: We thank the reviewer for the comments regarding the manuscript and the value of our study. We agree that a third replicate would help to strengthen the conclusions. As advised, we have now performed a third biological replicate for both the acetylome study and the whole proteome analysis at all time points of infection. All the figures and tables have been revised to incorporate the new data. The new proteome and acetylome data were in full agreement with our initial observations, such as the increased acetylation within the mitochondria and at the nuclear periphery during infection. Additionally, given the stringency of our initial criteria for filtering identified acetylations prior to their quantification, the addition of the third replicate led to an increase in the number of acetylated peptides that could be quantified. Many of the previously discarded acetylation sites were now rescued if these were also detected in the new biological replicate. Specifically, the inclusion of a third biological replicate allowed us to use a filter in which acetylated peptides had to be present in 2 out of 3 replicates at 1% false discovery rate (FDR) and with CV less than 100% in at least one time point. Out of the identified 6180 acetylated peptides, 5756 passed this filtering. The quantified proteins also increased from 5180

in the initial two replicates to 5216 in the three replicates. This is now clarified in the manuscript (in results on page 6-7 and methods on page 32).

2. Please include description of how the reliability of the quantification measurements between biological/technical replicates was handled (statistical analyses, estimates of uncertainty, etc.)

>Response: We thank the reviewer for highlighting the need to include more detailed description of how reliability of quantification was handled. In the revised manuscript, each sample for the acetyl-IP and whole proteome datasets has now been analyzed in three biological replicates. To assess the reliability of label-free quantification, we calculated the percent coefficient of variation (%CV) between the biological triplicates for each time point. For the acetyl-IP data, individual acetyl-peptide abundances were discarded if the %CV was ≥ 100 in all time points. For the remaining values, a %CV histogram was constructed using values across all time points, demonstrating that 90% of the data had a %CV of $\leq 100\%$. We have performed the same %CV analysis of the whole proteome dataset, showing that 95% of the data had a %CV of $\leq 54\%$. These ranges of %CV values are typical for PTM enrichment studies and whole proteome profiling measurements, the latter of which usually has lower variability. These histograms can be found in Supplementary Fig. 3 for acetylome data and Supplementary Fig. 2 for proteome data, and the individual CV values are now also listed in the Supplementary Tables 1 and 2. The text was also revised to indicate this (page 7 in results and pages 30-32 in methods).

3. I have tried my best to download the MS proteomics data using the accession number PXD009839 from the ProteomeXchange Consortium, however, no projects found for search term PXD009839. Without having access to their raw data or peak lists I cannot tell the quality of their MS experiments.

>Response: We are sorry that the login information did not work. We initially tested the link and it worked. Since we have now added a third replicate, we made a new deposition of the entire data, which can be accessed at ProteomeXchange Consortium with the following reviewer login information:

Username: reviewer60717@ebi.ac.uk

Password: rRJspYAf

4. The K to Q mutations of LMNB1 mimic constitutively acetylated state while conserving the same charge, can the authors also create K to R mutants which will mimic non-acetylated state of LMNB1? This would nicely support the roles of the acetylation sites analyzed.

>Response: We followed the reviewer's advice and have now generated K-to-R mutants for sites 109, 134, and 483, and assessed the effect on viral titers (new Fig. 4f-h). This further highlighted the relevance of the K134 site, as the mutation of the other sites did not impact virus titers (new Fig 4f-g).

> We now further expanded our analyses of the K134 site to more completely compare the Q and R mutants (revised Fig. 4 and 5, Supplementary Fig. 9 and 10). The K134R mutation had an intermediate phenotype, leading to a decrease in virus that was not as pronounced as for

K134Q (new Fig. 4h). This impact on virus titer can derive from a number of factors, including a difference in the number of released viral capsids from the nucleus, a disruption in the assembly of viral particles, or a reduction in the overall infectivity of the produced particles. To start to address this, we also measured cell-associated virus. K134Q expressing cells showed a significant reduction in cell-associated virus, while K134R had an intermediate phenotype, which was not significant when compared to cells expressing WT LMNB1. This suggests impaired capsid nuclear egress can in part contribute to the observed reduction in the virus titers for K134Q.

> In light of these new results, we now expanded our microscopy analyses of viral capsid nuclear egress to compare cells expressing WT, K134Q and K134R (Fig. 5a-d). Late in infection (120 hpi), slightly more viral capsids were accumulated in the nucleus in the acetyl-mimic mutant (K134Q) cells in comparison to both WT and K134R LMNB1 (Fig. 5b-c), suggesting an inhibition of nuclear capsid egress. Additional images of 120 hpi are shown as both single slice and maximum projection in Supplementary Fig. 9. In assessing the 3D volume of the pUL32-GFP particles inside the nucleus, we observed that K134Q cells displayed an accumulation of aggregated particles when compared to both WT and K134R cells (Fig. 5d). Altogether, these observations suggest that the phenotype is likely not driven by an acetylation-ubiquitination switch at K134, but rather a specific impact of acetylation (or the change in the charge of the side chain at that position) on the lamina.

> An interesting observation was that, at an intermediate time point (96 hpi), nuclear capsid egress seemed to progress more effectively in K134R cells (than in K134Q and WT LMNB1 cells) (Fig. 5b-c); this difference was only temporary, and capsid egress was similar in WT and K134R cells later in infection. However, this may further highlight the importance of this site in lamin regulation. Therefore, we further investigated the lamin morphology in the presence of these mutations at late time points of infection (96 and 120 hpi, Fig. 5e and Supplementary Fig. 10). The cells expressing WT or K134R LMNB1 displayed the known disruption of the nuclear periphery late in infection, with infoldings of the lamina and the formation of a kidney-shape nucleus. The curvature of this kidney shape is known to accommodate the viral assembly complex (as illustrated in Fig. 1a and shown Fig. 5a). In contrast, the cells expressing K134Q displayed a significant reduction in the disruption of the nuclear periphery, with less infoldings observed (Fig. 5e-f). Kidney-shaped nuclei were visible, but had a less pronounced aspect than the WT and K134R LMNB1 cells (Fig. 5e). Additional cells that display all the observed phenotypes are shown in Supplementary Fig 10. These results demonstrate that K134 acetylation inhibits the disruption of the nuclear lamina.

> The description of these new results has been now included in our manuscript (page 15-17), and we have also expanded our discussion (pages 20-22).

5. Can anything be said about the relative stoichiometry of the K-acetylation found? For example, what is the ratio of K109 acetylation in K134Q/K483Q mutants or the ratio of K134 acetylation in the other two mutants? I assume the relative level of modification is important to support the functional results for LMNB1, which may affect nuclear egress of viral capsids.

>Response: The reviewer brings up a nice point that we are also interested in. Since the detection of acetylation in our workflow is based on immunoaffinity enrichment of acetylated peptides, we do not obtain information about the relative stoichiometry of the identified

acetylations. We have extensively tried to isolate endogenous LMNB1 using commercially available antibodies to quantify the relative levels of these modifications. However, these experiments were not successful, as the antibodies did not have enough affinity for effective protein isolation. Therefore, although gaining a global understanding of the impact of different modifications and their relative stoichiometry on LMNB1 functions is of interest, it is outside the scope of this study. For future studies, the relative modification levels are not only of interest among different acetylation sites, but also between acetylations and other possible posttranslational modifications. For example, phosphorylations of LMNA/C was shown to trigger localized disruption of the lamin, and to be involved in lamin dynamics during mitosis. The same serines are also conserved in LMNB1 (new Supplementary Fig. 8, serine residues marked in blue), but it remains to be determined whether these sites can also be phosphorylated. It is possible that the K134 acetylation also provides a local regulation of the lamin stability and dynamics. Of note, the same lysine residue was previously reported to be ubiquitinated, and LMNB1 is known to be targeted for degradation upon ubiquitination. However, it is not yet known whether this specific site initiates its degradation, and this modification was so far only studied in a different biological context and not during infection. Our K134R mutant studies suggest that it is not an ubiquitination event that is needed for the infection-induced local disruption of LMNB1. However, our results demonstrate that this residue plays an important role in the flexibility of the lamin, with acetylation restricting the ability of the virus to induce the disruption of the nuclear periphery. It is possible that lysine ubiquitination and acetylation, as well as serine phosphorylation on LMNB1 and LMNA/C are dynamically regulated during infection. As the exit of the virus capsid from the nucleus to the cytoplasm requires only the local disruption of the lamina, and not global for the entire lamin, these modifications are not expected to have high stoichiometry. It is noteworthy that the previously detected phosphorylations on LMNA/C are present on the tails and inter-domain regions, while several acetylations on LMNB1 and LMNA/C that we identified are present in the coils of the lamin rod-domains. To address the reviewer's comment, we have now: 1) included a discussion of the stoichiometry and possible cross-talk of lamin PTMs (page 21-22), as well as 2) a new figure panel showing the conservation of these sites among LMNB1 from different species (see new Fig. 4c, Supplementary Fig. 8, and answer to point 6 below).

6. Is there a plausible explanation to describe why K109/483 do not have the same functional importance, yet these sites are still modified?

>Response: We thank the reviewer for highlighting the distinction between site-specific functions. Posttranslational modifications are known to regulate an array of protein properties, such as protein folding, stability, activity, interactions, and signaling. For the moment, our results show that the K134 site is relevant for viral infection. It is possible that the other acetylated sites have different functions that do not directly impact viral titers. For example, the K109, K261 and K483 sites are evolutionarily conserved across vertebrates, while the K134 site seems to have evolved in mammalian species (Fig. 4c). Therefore, it is possible that the K109, K261 and K483 sites are involved in the structural regulation of lamin or in interactions with other cellular factors that function in distinct pathways. For example, K483 falls in the Ig-like fold region, which is known to be involved in interactions with non-lamina proteins. We have now added a discussion regarding how modifications can differentially impact protein function (page 21-22).

7. The specific criteria in the confidence determination for peptide modification need to be explained in this manuscript.

>Response: The ptmRS node in Proteome Discoverer was used to calculate the probability that the acetyl modification is present on each potential lysine residue within the peptide. Based on the observed fragment ions, each potential site is assigned a localization probability, which is translated into a percent confidence score. A minimum score threshold of 75% confidence was required, which removed modified peptides where the acetyl modification could not be confidently assigned. We thank the reviewer for pointing out this missing information, and we updated the explanation in the Methods section (page 32).

8. Page 7, line 135: Please include the detailed description of “ppm”.

>Response: We thank the reviewer for highlighting the confusion surrounding “ppm” in this context. In order to assess the number of acetylation sites that we detected relative to protein abundance, we used the average protein abundance in parts per million (ppm) sourced from PaxDB. Proteins in this database are scaled in ppm relative to the whole proteome of the reference dataset, which in this case was the dataset: *H. sapiens*- Whole organism (integrated). We have included a description of this approach in both the legend of Figure 1 and in the Methods (page 33) and defined the term in the text (page 7). As not all the proteins found to be acetylated in our acetyl-IP analyses were also detected in our whole proteome study (especially for some low abundance proteins), we opted to use the abundances from PaxDB (Figure 1c-d). However, we have now complimented this assessment of protein abundance with an Intensity Based Absolute Quantification (iBAQ) analysis, which used the quantitative information from our proteome analysis. Normalized protein abundances from our whole proteome analysis (summed precursor intensities) generated by PD v2.2 were divided by the number of theoretically observable tryptic peptides (8-30 amino acids in length, generated by Perseus). iBAQ abundances across infection time points were averaged to represent the average protein abundances during infection. We have included this complimentary data in Supplementary Figure 1f, and are mentioning it in the results section on page 8.

Reviewer #2 (Remarks to the Author):

Murray et al provide an interesting study and resource making the first systematic analysis of changes in acetylation of viral and host proteins throughout infection with human cytomegalovirus (HCMV). There is increasing evidence that lysine acetylation is an important regulatory PTM that can influence a multitude of host responses, and can have roles in innate immunity. Their study is also interesting since relatively little is known about acetylation of viral proteins in general, and very little is known about acetylation of HCMV proteins.

The authors describe an approach to measure changes in acetylation that sensibly includes correction for overall changes in expression of proteins at a whole cell level. They further attempt to define acetylation over time on a subcellular basis, using results of their previous

proteomic study that assigned proteins to subcellular structures based on density gradient fractionation (Beltran et al Cell Systems 2016). Finally, they find interesting new acetylation sites in nuclear lamins, and provide some functional data that relates these modifications to viral replication via inhibition of capsid egress. They also identify K203Q acetylation of HCMV UL26 as an important PTM that modulates viral replication. The study is in general well performed, and the manuscript well written. However, I have several concerns that should be addressed, under ‘major points’ and ‘minor points’ below:

Major points

1. The number of proteins quantified is relatively modest, due to the lack of peptide fractionation. In their whole cell lysate analysis, only 1065 proteins were quantified (Table S1). Had peptide or protein pre-fractionation been performed (for example by running lysates onto a gel and cutting into 6 fractions, as per their 2016 study), many more proteins would have been quantified. Similarly, the number of peptides in their acetyl analysis would have increased proportionally. For the latter, it may be that this would have presented practical difficulties due to the sheer amount of material required; however this should at the least be explicitly discussed, and a detailed table provided showing numbers of proteins and peptides quantified in each part of each experiment (i.e. each time point in duplicate for the whole cell lysate analysis, the overlap to generate the protein table S1, and the overlap between experiments – e.g. between whole cell lysate and acetyl enrichment).

>Response: We thank the reviewer for the comment, and apologize for the likely misunderstanding. We initially quantified 5180 proteins in our initial proteome dataset, requiring a minimum of 2 peptides identified per protein and 1% false discovery rate (FDR). The 1065 referred only to the previous subset of proteins that were also found to be acetylated in our acetylome study from the first two replicates. To remove this misunderstanding, we now reworded our description in the text and table legends to clarify the numbers of proteins or acetylated peptides quantified from the proteome and acetylome analyses, respectively.

> However, to further address the reviewer’s comment, we now improved the depth of analysis in the revised manuscript by performing a third biological replicate for both the whole proteome analysis and the acetylome study at all time points of infection. All the figures and tables have been revised to incorporate the new data (i.e., Fig. 1c-e, Fig. 2a-e, Fig. 3 a, d, Fig. 4a-b, and Fig. 6a). The new proteome and acetylome data are in full agreement with our initial observations, such as the increased acetylation within the mitochondria and at the nuclear periphery during infection. Given the stringency of our initial criteria for filtering identified proteins and acetylations prior to their quantification, the addition of the third replicate led to an increase in the resulting corresponding numbers. Many of the previously discarded acetylation peptides were now rescued if these were also detected in the new biological replicate. Specifically, the inclusion of a third biological replicate allowed us to use a filter in which acetylated peptides had to be present in 2 out of 3 replicates with CV less than 100% in at least one time point. Out of the identified 6180 acetylated peptides, 5756 passed this filtering. This corresponds to 2018 identified acetylated proteins, with 1816 acetylated proteins passing our filtering for quantification. The quantified proteins in the whole proteome analysis also increased from 5180 in the initial two replicates to 5216 in the three replicates. This is now detailed in the text of the corresponding results section (page 6-8).

> To provide further detail about the numbers of identified acetylations and proteins in our studies, we followed the reviewer's advice and now also include supplementary figures for: (1) the number of host (Supplementary Fig. 2c) and viral (Supplementary Fig. 11b) proteins identified at each time point in each replicate, and (2) the number of acetylated peptides quantified at each time point in each replicate, as well as the corresponding number of acetylated host (Supplementary Fig. 4a) and virus (Supplementary Fig. 11a) proteins.

> We have also added a new supplementary table (Supplementary Table 1) that lists all the host (Table S1A) and viral (Table S1B) proteins identified in each time point for each biological replicate of the proteome analysis. We have added a column that indicates the overlap between this proteome dataset and our acetylome dataset, i.e., if a protein was detected in both the proteome and acetylome datasets.

> In Supplementary Table 2, we show the host acetylated peptides that passed our filtering criteria for use in our quantitative analyses (i.e., present in 2 out of 3 replicates, 1% FDR, and CV%<100 in at least one time point). Analogous information is provided for the acetylated viral peptides (Supplementary Table 5A).

> For further clarity, we have also included as supplementary figure panels a series of Venn diagrams displaying the overlap between replicates at each time point for both the host acetylated peptide dataset (displaying only acetylated peptides that passed our filtering and were subsequently used for quantification) (Supplementary Fig. 3a) and the host whole proteome dataset (Supplementary Fig. 2a).

> Finally, to address the reviewer's comment regarding comparison of our dataset with previous studies, we are now also providing a comparison between our proteome results and the most in-depth proteome study during HCMV infection to date, published by Weekes et al. Cell 2014 (Supplementary Fig. 2d-e). This analysis shows that the results are fairly comparable, with the majority of the proteins being identified in both studies. If data analysis is performed using the criteria used in our current paper, which required a protein to be detected with a minimum of 2 unique peptides and 1% false discovery rate (FDR), and isoforms were not counted separately, then our study identified 5216 proteins, while the Weekes et al. study identified 6276 (Supplementary Fig. 2e). If we use the criteria described in the Weekes paper (≥ 1 unique peptide per protein, 1% FDR), then the identifications are 6411 for our study and 7408 for the Weekes study, without isoforms (Supplementary Fig. 2d). Of course, differences exist between the two studies, starting with the use of different viral strains (AD169 in our study and Merlin in the Weekes paper) and, just as important, in the experimental workflow. Weekes et al. digested proteins from whole cell lysates by LysC and Trypsin, and peptides were separated into 12 fractions, and this approach nicely enhanced protein identification. Despite these differences, the overlap between the identifications in these two studies is high, with over 75% of the proteins identified in the Weekes study being also detected in our study, and ~90% of our proteins being identified in the Weekes study. In summary, as we analyzed the proteome only as a normalization tool for our acetylome study, and most of our identified acetylated proteins were quantified at the proteome level by this approach, the depth of our whole cell proteome is adequate for our acetylation-focused study.

2. The authors employ 'imputed values' to account for missing values, where a protein is not quantified in every analysis. These appear to make a minor contribution to the WCL analysis, however are more important in their acetyl study. I would suggest (a) providing a supplementary

table showing the number of imputed values per experiment as a proportion of total, and (b) in Table S2, highlighting in columns T-AC which values are imputed (e.g. in colour). I realise this is shown in AD-AM by cells lacking data, however it is important for the use of this data by others to recognise which values were not actually measured.

>Response: We thank the reviewer for highlighting the need for clarity with respect to the imputed values. We should clarify that the imputed values were only used for our broad-spectrum analyses of overall trends (Fig. 2b, Fig. 3a, Supplementary Fig. 4b-e), as imputing missing values facilitated clustering and assessment of changes relative to uninfected cells. The %CV values were calculated prior to imputation. Also, as the reviewer correctly indicated, figures displaying site-specific changes on certain proteins did not include any imputations, and missing values (not detected) are represented in white. As suggested by the reviewer, we have now (1) included a table with the percentage of total quantitative values that were imputed for each sample (Supplementary Fig. 4a), and (2) highlighted the values in columns V-AJ of Supplementary Table 2 that are imputed. The percentage of imputed values for the 24 hpi sample in replicate two is higher compared to the other samples because of greater sample processing losses. Consequently, as fewer peptides were identified in that run, there were more missing values. Overall, to mitigate the impact of imputation, we have performed a third biological replicate to improve the reliability of our acetylation trend analyses. Finally, given the potential biological relevance of missing values over the time course of infection, we have updated the figure legends to further clarify which figures display data with missing values displayed as blanks (Fig. 2c-e, Fig. 4a-b, Supplementary Fig. 7a).

3. The normalisation to absolute whole cell lysate protein abundance is important, however can potentially be confounded by proteins that were only measured by single peptides, or by the use of imputed values for protein-level measurements. This should also be discussed, and a column added to Table S2 to reflect the number of peptides measured per protein at a whole cell level. Where imputed whole cell values were used should also be shown.

Response: We thank the reviewer for indicating that this was not previously clearly explained in the methods. We have significantly revised the methods section with stepwise descriptions of the LFQ workflows for both acetyl-peptide and total proteome analyses (page 30-32). For the total proteome measurements, for a protein to be considered identified, it had to be detected with at least two unique peptides. Additionally, imputation of abundance values was performed only at the peptide level, when two out of three values were measured and using a replicate-based imputation method, which imputes a single missing value from a random value selected around the median of the replicates. Peptide ratios are calculated as all pairwise combinations of respective replicate abundances, and the protein ratios are calculated as the median of the peptide ratios. Using this pairwise ratio approach, the negative effects of imputation, i.e. effect of increased variability, are significantly mitigated. Referring to the issue of protein level imputation, we did not impute protein-level LFQ directly; so, if no peptides were detected in a specific condition, the value was left as null. The corresponding ratios involving this value were either infinity (arbitrarily set at 1000) or null, and were not used for acetyl-peptide normalization. For greater transparency, as requested we have included a column in Supplementary Table 1 to indicate the number of peptides used for quantification at the whole cell level.

4. The authors derive confidence in their whole-cell data from an analysis of up- or down-regulated host pathways. However, this could be performed in greater depth and precision by comparing their current temporal whole-cell data to their previous 2016 published whole cell data set. From the cluster plots, their dynamic range of changes even by 96 h of infection seems quite modest (from Table S1, only ~20 proteins up or down-regulated >4-fold) - was this seen before?

>Response: We appreciate the comment, which is connected to the comment 1 above. As indicated above, we regret the misunderstanding and have now included additional tables and data comparisons to demonstrate the adequate depth of our proteome data, and its similarity to the most in-depth previously published HCMV infection proteome, published by Weekes et al. As the 2016 study published by Jean Beltran et al. focused on assigning proteins to different organelles during infection, and identifying protein movement between organelles, the study by Weekes et al (Cell 2014) was more similar to the whole-cell proteome dataset in our current study. As indicated in response to point 1, when comparing our proteome data with that from the Weekes study, the overlap between the two studies is high, with over 75% of the proteins identified in the Weekes study being also detected in our study, and ~90% of our proteins being identified in the Weekes study. Additionally, the changes in protein abundances mirror those previously observed. For example, the number of proteins found to be upregulated over two fold in our study nicely match the numbers reported in the Weekes paper. Specifically, the numbers of upregulated proteins (> 2 fold) are 140 versus 162 at 24 hours post infection (hpi) in our study when compared to the Weekes study. Similarly, the numbers are 269 versus 398 at 48 hpi, 407 versus 539 at 72 hpi, and 488 versus 575 at 96 hpi. Furthermore, the proteins that undergo changes in abundances match the prior knowledge from biological studies on HCMV infection, such as the highly elevated immune response. Therefore, our observations from the proteome analyses are consistent with the previous literature. Supplementary Fig. 2d-e shows the good overlap between the proteins identified in these studies, and main Fig. 2a displays the expected changes in protein abundances, which match prior knowledge from the literature.

5. “To determine temporal changes in acetylation, we focused on identifications that were consistent between biological replicates at the different time points, and for which we could accurately determine levels” – please explain this statement, and justify in detail any data that has been discarded in their analyses.

> Response: Given the use of primary human fibroblasts for our study, the infection with HCMV that has a slow replication cycle (over 4 days), and the necessity of large amounts of samples for performing acetylome enrichments at all different time points of infection, we initially used 2 biological replicates for our acetylome study. However, since we wanted the quantification to be robust, we were stringent in what we considered suitable for quantification. The acetylated peptides had to be present in both replicates at 1% false discovery rate (FDR) for any given time point, with CV less than 100%. This indeed led to many identifications not being included in the previous quantification. As we now performed a third biological replicate for both the acetylome and proteome studies at all time points, this allowed us to use a filter in

which acetylated peptides had to be present in 2 out of 3 replicates at 1% false discovery rate (FDR), and with CV less than 100% per time point. Out of the identified 6180 acetylated peptides, 5764 passed this filtering. We have now clarified this in the manuscript (page 8).

6. “Furthermore, by monitoring in parallel the corresponding protein abundance levels (Fig. 2b, left) and normalized acetylation versus protein levels (middle), we found that a majority of the identified alterations in site-specific acetylations are independent of protein abundance changes.” Comparing the left to right hand panels of Figure 2b, it seems that many of the upregulated acetylated peptides are also upregulated at the whole protein level, and a proportion of the downregulated peptides are similarly downregulated. To justify their statement, some form of quantitation and statistics should be provided (a series of dot plots for each time point might help). Specific proteins (e.g. IFITs etc) could then be indicated on these plots, which would make this much easier to understand.

>Response: We agree with the reviewer that some of the acetylated peptides that have increases or decreases in abundance also display a similar trend at the protein level. For some of these, even after normalization to the changes in protein levels, the acetylations still further increased or decreased. To more clearly display how the trends in acetylated peptide abundance are (or are not) maintained upon normalization to protein abundance (middle column of Fig. 2b), we have added dot plots as per the reviewer’s suggestion (Supplementary Fig. 4b-e). The dot plots show the log₂ fold changes in acetylated peptide abundance/mock versus the log₂ fold change in protein abundance/mock for each time point. As suggested, we have indicated specific protein acetylation sites of interest on these plots. This analysis supports that, although some changes in protein acetylation match the changes in protein abundance, the alteration in protein abundance is not the main driver of the observed changes in acetylation levels. Specifically, the r^2 values are below 0.3 for all time points when comparing acetylation levels with protein abundance levels.

7. “Strikingly, nearly all of the sites on IFIT2 decline in abundance compared to their levels at 24 hpi. This points to a possible rapid acetylation-dependent IFIT2 response that is specific to the early stages of infection.” I disagree – their time-course profiles of whole protein IFITs 1,2,3 also decline. It would be useful to include time-course plots showing (a) whole protein abundance (ALL peptides – already excluding modified peptides) and (b) acetylated peptide abundance. If their assertion is at all justified, there should be a statistically significant difference between both groups.

>Response: From our initial two replicates of proteome analyses, we observed large increases in the protein abundances of IFIT1, IFIT2, and IFIT3, ranging from 16 to 128 fold increases compared to mock cells. Consistent with their biological functions, these increases are observed relatively early in infection, after which the protein levels reach a plateau or slightly decline, although remaining still substantially elevated compared to mock. This large increase in IFIT protein abundance levels was now reinforced by our third replicate, as log₂ fold changes pointed to similar increases (Supplementary Table 1). To further clarify this, as suggested by the reviewer, we now include time-course plots displaying the increase in protein abundances (Supplementary Fig. 5b-d, blue lines).

> Most of the acetylated IFIT peptides were only detected upon infection (Fig. 2c, red boxes). The addition of a third replicate led now to the identification of a few acetylated peptides in uninfected cells (i.e., detected in 2 out of the 3 replicates), and for these peptides we could therefore now normalize their relative abundances during infection to uninfected cells (Fig. 2c, black boxes). The results support our previous interpretation that this family of IFIT proteins illustrates an exception to the independence of protein acetylation levels from protein abundance. Specifically, both the protein and acetylation levels increase; however, the increase in acetylation does not always reach the level of the protein. We have now clarified this in the text (page 9), and have removed any confusing interpretation regarding IFIT2.

8. Their conclusions about the correlation between site-specific acetylations and organellar translocations seem at the least highly speculative. For example, MGST3 apparently translocates from the mitochondria to the peroxisome at 48 h of infection then back again. Furthermore although there is no change in acetylation upon the initial translocation, acetylation decreases for the peroxisome to mitochondrion shift. The authors show this and several other similar examples (which presumably represent their best data). They speculate “It remains to be seen whether acetylation drives the changes in localization or whether the localization induces a change in acetylation.” In fact neither may be the case. Since changes in peptide acetylation within any given organelle hasn’t actually been measured, it’s impossible to know whether the acetylated form of the protein translocates, with the non-acetylated form of the protein staying put, or if the changes in acetylation are actually completely irrelevant to organellar location. The authors could potentially re-search the data from their 2016 publication with a variable acetyl modification. Although there was no acetyl-specific enrichment, one might expect to identify some acetylated peptides anyway. This might (or might not) help to support their conclusions. Alternatively, if any anti-acetyl antibodies are available for any of the proteins of interest, immunofluorescence could provide supportive information. Furthermore, could some measure of the reliability of their assessment of protein localisation / movement (from their previous publication) for these examples be provided in the supplementary (e.g. all the peptide profiles from each protein)? A protein translocating at a single time point between mitochondrion and peroxisome or Lysosome and ER/Golgi seems surprising and gives concern that the measurement of location at this time point may not be correct. Without any supportive data, the conclusions from this section, particularly “By integrating our acetylome dataset with spatial-temporal information of protein localization during HCMV infection, we provide a perspective of how site-specific, temporal acetylation can be regulated at a subcellular level to impact viral replication” are a very considerable overreach.

>Response: We thank the reviewer for the comment, and we have added both the suggested new analyses and text changes to carefully address this point.

> As clarification, this analysis was meant to provide information about the usual subcellular localization of the proteins that we detected as acetylated. For example, many well-documented mitochondrial proteins are observed in our study to have increased acetylation levels. This is why this results section mainly focused on describing the changes observed within certain organelles, in particular in the mitochondria and the nucleus. For completeness and transparency, we wanted to also indicate that some of the proteins either known or predicted to change localization during infection were also found to be acetylated in our study. We

specifically did not want to state that the acetylation is responsible for the translocation, and this is why we listed some of the possible interpretations. We just wanted to ensure that the readers have this information, as it may impact the function of those proteins. However, we fully understand the point of the reviewer, and we carefully adjusted the text (pages 13-14) to avoid misunderstanding, and have performed additional analyses to support some of these organelle-specific localizations.

> As we could not find antibodies against the specific acetyl sites of interest, we followed the reviewer's suggestion and reanalyzed the density fractionated proteome data from Jean Beltran et al. 2016 to search for acetylated proteins. As the reviewer predicted, only few acetylation sites were detected from that dataset, given the lack of acetyl enrichment. However, we did detect proteins that were acetylated at the same sites in both Jean Beltran et al. and in our current acetylome study. Many of these proteins were organelle resident proteins that do not change localization during infection. These proteins, acetylation sites, and their subcellular localization are now shown in a new Supplementary Table 4. For these proteins, the localization during infection was determined from Jean Beltran et al. 2016. As the reviewer suggested, their localization scores from Jean Beltran et al. 2016 are now provided in the supplementary table. These localizations were also consistent with their allocation in Uniprot and Protein Atlas, and we indicate this in the table. Therefore, this comparison supports that the proteins localized within those subcellular compartments are acetylated.

> Another subset of acetylations common between our acetylome study and the Jean Beltran et al. dataset was on proteins predicted to translocate in the Jean Beltran et al. study. As the reviewer nicely predicted, this serves to support the possible acetylation of translocating proteins, and we now list these in new Supplementary Table 4. As the reviewer suggested, their localization scores from Jean Beltran et al. 2016 are also provided. Furthermore, these proteins are also known to translocate or to have multiple localizations outside the context of infection, and we indicate this in the supplementary table. For completeness, Fig. 3b shows the total number of proteins found in our study to be acetylated and previously predicted to translocate during HCMV infection in fibroblasts by Jean Beltran et al. 2016. Again, some of these proteins are also known to translocate in other biological contexts (e.g., VAPA). To further address the reviewer's comment, we also streamlined this paragraph. The previous examples that were originally included in the manuscript were not meant to show our best data, but rather highlight some interesting novel findings that may be relevant to HCMV infection. To avoid the feeling of overinterpretation, we now removed these and replaced them with just two brief examples of proteins that are already well established to translocate, MYH9 (text) and VAPA (text and figure) (pages 13-14).

9. Although the findings with regards to the K132Q mutant are very interesting (200 fold drop in viral titre), the conclusions on mechanism (Figure 5b-c) are dubious. The only differences in UL32-GFP intensity in the nucleus they see are relatively modest changes at 120 h post infection – by which point infected fibroblasts if still alive would be falling apart due to the terminal stage of viral infection. Further data to better justify their mechanistic conclusions are warranted.

>Response: We thank the reviewer for the comment and have now added additional experiments to address potential mechanisms. To extend our analysis of the role of acetylation at K134, we have now generated a non-acetylatable charge-mimic R mutant for K134. For completeness, we

also generated R mutants for the other two investigated sites, K109 and K483. The mutations of K109 and K483 did not significantly impact virus titers (new panels Fig. 4f-g). The K134R mutation had an intermediate phenotype, leading to a decrease in virus that was not as pronounced as for K134Q (new Fig. 4h). This impact on virus titer can derive from a number of factors, including a difference in the number of released viral capsids from the nucleus, a disruption in the assembly of viral particles, or a reduction in the overall infectivity of the produced particles. To start to address this, we also measured cell-associated virus. K134Q expressing cells showed a significant reduction in cell-associated virus, while K134R had an intermediate phenotype, which was not significant when compared to cells expressing WT LMNB1. This suggests impaired capsid nuclear egress can in part contribute to the observed reduction in the virus titers for K134Q.

> In light of these new results, we now expanded our microscopy analyses of viral capsid nuclear egress to compare cells expressing WT, K134Q and K134R (Fig. 5a-d). Late in infection (120 hpi), slightly more viral capsids were accumulated in the nucleus in the acetyl-mimic mutant (K134Q) cells in comparison to both WT and K134R LMNB1 (Fig. 5b-c), suggesting an inhibition of nuclear capsid egress. Additional images of 120 hpi are shown as both single slice and maximum projection in Supplementary Fig. 9. In assessing the 3D volume of the pUL32-GFP particles inside the nucleus, we observed that K134Q cells displayed an accumulation of aggregated particles when compared to both WT and K134R cells (Fig. 5d). Altogether, these observations suggest that the phenotype is likely not driven by an acetylation-ubiquitination switch at K134, but rather a specific impact of acetylation (or the change in the charge of the side chain at that position) on the lamina.

> An interesting observation was that, at an intermediate time point (96 hpi), nuclear capsid egress seemed to progress more effectively in K134R cells (than in K134Q and WT LMNB1 cells) (Fig. 5b-c); this difference was only temporary, and capsid egress was similar in WT and K134R cells later in infection. However, this may further highlight the importance of this site in lamin regulation. Therefore, we further investigated the lamin morphology in the presence of these mutations at late time points of infection (96 and 120 hpi, Fig. 5e and Supplementary Fig. 10). The cells expressing WT or K134R LMNB1 displayed the known disruption of the nuclear periphery late in infection, with infoldings of the lamina and the formation of a kidney-shape nucleus. The curvature of this kidney shape is known to accommodate the viral assembly complex (as illustrated in Fig. 1a and shown Fig. 5a). In contrast, the cells expressing K134Q displayed a significant reduction in the disruption of the nuclear periphery, with less infoldings observed (Fig. 5e-f). Kidney-shaped nuclei were visible, but had a less pronounced aspect than the WT and K134R LMNB1 cells (Fig. 5e). Additional cells that display all the observed phenotypes are shown in Supplementary Fig 10. Altogether, our findings demonstrate that K134 acetylation inhibits the disruption of the nuclear lamina.

> The description of these new results has been now included in our manuscript (pages 15-17), and we have also expanded our discussion (pages 20-22).

10. This study employs the laboratory adapted HCMV strain AD169. Although this is convenient to work with and can easily be grown to high titre, it is less similar to a wild-type strain than less extensively passaged HCMV isolates, particularly as it lacks ~20 genes in the UL/b' locus. The authors should at least acknowledge this as a potential caveat.

>Response: We appreciate the reviewer's comment. As many of the molecular details of HCMV replication have been previously deciphered using infection with HCMV AD169 or Towne laboratory strains in fibroblasts, we selected this approach to provide a first foundation of knowledge of protein acetylation during HCMV infection in a context that can be compared to these previous studies. The HCMV AD169 strain that has been BAC cloned (Yu et al. from Shenk group, J. Virol. 2002) and well characterized has allowed us to study the different stages of the HCMV replication cycle, and the impact of infection on host protein acetylation. However, we fully agree with the reviewer that laboratory strains have certain limitations. Clinical strains, such as FIX, TB40/E and Merlin, also present both advantages and limitations, and their use in future studies would allow expansion of this knowledge to further interrogate the function of acetylation in both the context of active replication and latency. Another consideration is the remarkable broad tropism of HCMV and the possible cell-type specific response, as the virus can infect diverse tissues and cell types, including epithelial and endothelial cells, fibroblasts and smooth muscle cells. Future work addressing the function of cellular and viral acetylation in different cellular systems and during infection with clinical strains would further enhance the current understanding of HCMV biology and cellular pathology. As advised by the reviewer, we have now mentioned this in our discussion (page 26).

11. A number of the inferences in the text seem overstated – e.g. “This points to a likely major regulation of metabolic pathways during infection”; “at least 5% of the lysine residues encoded by the HCMV genome are acetylated, providing evidence that viral protein acetylation is likely a prominent regulatory event.”

>Response: We have adjusted the text to avoid giving the impression of overstatements. Just to clarify, as we have previously indicated in the manuscript, it has been elegantly documented by several groups that HCMV infection triggers a substantial regulation of metabolic pathways. For example, the flux through glycolysis is highly elevated, as well as fatty acid synthesis, and glutamine becomes the main carbon source for the TCA cycle. Our finding that acetylation on mitochondrial resident proteins is elevated during infection is very interesting in this context, as it indicates possible means for the modulation of these pathways. For example, increased acetylation is known to be detrimental to oxidative and intermediary metabolism. So, this statement was simply meant to highlight this major regulation of cellular metabolic pathways and as a transition to this background knowledge.

> Similarly, we apologize for not being explicit enough about what was meant in the sentence regarding HCMV acetylation. Our results indicate that at least 5% of the lysine residues encoded by the HCMV genome are acetylated. Although acetylation sites are likely to still be uncovered for viruses, human and other species, so far the most in-depth analysis of acetylation in human cells has shown that approximately 2% of lysine residues encoded in the human genome are acetylated (Svinkina T, et al. MCP 2015 from the group of Steve Carr). Therefore, the degree of HCMV protein acetylation seems to match and even surpass that known so far for human proteins.

> As the reviewer suggested, we have now reworded all the above-mentioned sentences to clarify these aspects (pages 12 and 18).

Minor comments:

1. For Table S1, please explain in the legend the meaning of ‘high’, ‘peak found’ and ‘not found’.

>Response: We now include a description of their meaning in the legend for Supplementary Table 1. Briefly, these terms are part of the output from Protein Discoverer (PD) v2.2, and refer to the peptides that were used for protein quantification. The terminology from PD v2.2 uses “High” to indicate that at least one unmodified peptide from that protein was sequenced by MS/MS in that particular sample. “Peak found” indicates that the peptides in that particular sample were found by feature matching to at least one other sample in which the peptides were confirmed by MS/MS sequencing. “Not Found” indicates that no peptides were detected for that protein in that sample. A detailed description of protein quantification can be found the Materials and Methods (page 31).

2. Figure 3a – the cluster plots should be plotted on the same scale (at least for the acetylated peptide / protein vs acetylated peptide; I realise different scales may be required for each organelle).

>Response: We apologize that the scales were interpreted as being plotted separately. The cluster plots were plotted on the same scale. Specifically, these scales were plotted -1.5 to 1.5, and the max and min for each were shown. To avoid confusion, we have now labelled the scale bar to indicate that all are plotted on the same scale (-1.5 to 1.5 log₂ fold change) (Fig. 3a).

3. Figure 3b – this is difficult to understand. Could this be more clearly explained in the legend?

>Response: Thank you for the suggestion. We have added to the description in the figure legend and included an example of how to interpret the boxes. The proteins along the center diagonal line are those observed or predicted to remain within their organelle during infection. As the Jean Beltran, et al (2016) study investigated the localization of proteins during HCMV infection in fibroblasts, the proteins determined in that study to remain within an organelle (even if their protein abundances changed or did not change) are shown in the light grey triangles along the middle line. The boxes displaced from the diagonal indicate whether the proteins that we found to be acetylated are either known or predicted to change localization during infection (from Jean Beltran, et al. 2016). For completeness and transparency, if some proteins detected in our acetylome study were not detected in the Jean Beltran, et al. 2016, then the localization of those proteins was obtained from Uniprot. Since we cannot state whether these proteins may or may not move during infection, we show those numbers in the dark grey triangles along the center diagonal.

4. Supplementary Fig 4b should read Supplementary Fig 3b line 305.

>Response: We thank the reviewer for identifying this error. Given the addition of new data and supplementary files, this data has become part of Fig. 5e.

5. Line 362 should read Fig. 6c not Fig. 6b

>Response: We thank the reviewer for identifying this error. The correct figure sub-panel is now referenced.

6. Their conclusions in Figure 1c comparing to average protein abundances from PaxDb could be improved by estimating actual protein abundances / cellular concentrations using e.g. the Histone Ruler (see Wisniewski et al), iBAQ etc.

>Response: We thank the reviewer for suggesting that we use iBAQ or the Histone Ruler to improve the representation of protein abundances shown in Fig. 1c. As the Histone Ruler would have required us to accurately determine the amount of cellular DNA present in the samples, we did not use this approach since HCMV genome replication would have skewed this data. Therefore, as the reviewer suggested, we turned to iBAQ and have now included a graph of averaged iBAQ abundances across infection time points in Supplementary Fig. 1f and an additional description in the text (page 8) and methods (page 33). The iBAQ approach uses the quantitative information from our own proteome analysis, nicely complementing our original PaxDB analysis. As not all the proteins found to be acetylated in our acetyl-IP analyses were also detected in our whole proteome study (especially for some low abundance proteins), for completeness we opted to also retain the analysis using abundances from PaxDB (Fig. 1c-d). Overall, the combination of the iBAQ data and the PaxDB data present a more complete picture of the number of acetylation sites that we detected on proteins with a wide range in abundance.

7. Figure 2a – please provide a supplementary table of all biological processes quantified in this analysis.

>Response: We thank the reviewer for indicating the relevance of this information. We have now included this information in Supplementary Table 1C. This data was generated through the Reactome GO Biological Processes Cytoscape plugin in a manner analogous to that which was used for the acetylated peptides. Related terms were aggregated into broader categories for display in Fig. 2a.

8. For the localisation analysis, please provide details of in which cases this was derived from their previous study, and which from Uniprot.

>Response: As suggested, we have added a column to Supplementary Table 3 to indicate the source of the localization data (column M). Additionally, we have added another column (column N) that shows whether the localization is consistent between the Jean Beltran 2016 paper and Uniprot. As the Uniprot information is not necessarily from infected cells, this comparison indicates whether a given localization is consistent for a particular protein. Furthermore, for the subset of acetylated proteins identified in both our acetylome study and the previous organelle proteome study (see response to main point 8 above), we now provide a new Supplementary Table 4, which also indicated whether the knowledge of localization derived from the Jean Beltran et al paper, Uniprot, or the Protein Atlas.

9. Please provide a username and password for review of the deposited RAW files.

>Response: We apologize that the login information that we initially provided did not work. We initially tested the link and it worked. Since we have now added a third replicate, we made a new deposition of the entire data, which can be accessed at ProteomeXchange Consortium with the following reviewer login information:

Username: reviewer60717@ebi.ac.uk

Password: rRJspYAf

> This is provided in the “Data availability” section of the methods.

10. Statistical analyses seem appropriate, apart from where extra analysis is needed as detailed above.

>Response: We sincerely thank the reviewer for the feedback, as we believe that all the suggested changes have made the paper stronger. As indicated above, we have now included a third replicate and have made all the changes suggested by the reviewer.

Reviewer #3 (Remarks to the Author):

In their current paper Murray et al use a mass-spectrometry screening approach to quantify the level of a common protein modification - lysine acetylation - during CMV infection of human fibroblasts (MRC5). Their approach is methodologically sound and their results are novel. While the manuscripts leans heavily towards the descriptive side, this is not surprising given the scarcity of research on protein acetylation. The major novelty of the paper is in the global identification of acetylation events on both the human and viral proteins, which are likely to open the door for more mechanistic investigations. The authors further elaborate on two specific findings; the acetylation on the host protein Lamin B and the acetylation of the viral protein pUL26.

For Lamin B, the authors identify 3 lysine residues for which acetylation increases during the course of infection. They overexpress acetylation-mimetic mutants of these sites (replacing the lysine with glutamine) and find that overexpression of one mutant (K134Q) leads to a 200-fold reduction in the number of infected cells.

For pUL26, the authors identify a single lysine acetylation (K203). They construct two mutant viruses - an acetylation-mimetic mutant, K203Q, and a charge-mimetic mutant, K203R and find that the acetylation-mimetic mutant is less fit, showing fewer infected cells, while the charge-mimetic is as fit as wild type.

Major concerns:

1. Viral titrations: The authors use IE1 staining and report the results as infectious units/ml. As infectious units usually refers to the number of plaque producing particles this is misleading. This is especially important in cases where the effect might be later than viral entry and IE expression, such as in the case of Lamin B possible effect on nuclear egress. The preferable approach would be the use of plaque assays to calculate the results as PFU/ml. Alternatively, the authors can rewrite the relevant paragraphs to say that they measure the effect on IE expression.

>Response: We apologize that this method was not explained sufficiently. The IE1 staining method that we used here involves i) infecting cells expressing the LMNB1 constructs, ii)

collecting virus from the supernatant at 120 hpi, iii) using this supernatant to infect a reporter plate of WT fibroblasts, and iv) staining for IE1. Therefore, the virus that is used to assess the impact of the LMNB1 mutation derives after a complete replication cycle. This method has been used as an alternative to plaque assays by numerous groups and shown to provide information regarding alterations at different stages of the virus replication cycle. This is because, similar to a plaque assay, a reporter plate is infected with virus deriving at the end of a replication cycle. We are now clarifying this in the text and methods (pages 15 and 39), and provide references to other studies that used this approach.

> Additionally, to further address the reviewer's comment, we have conducted a TCID50 assay to compare extracellular virus produced in LMNB1 WT and K134Q cells (new Supplementary Fig. 7b). These data support the reduction in viral titer observed in the presence of LMNB1 K134Q.

2. The role of Lamin B acetylation in CMV infections:

a. The fact that the same residue, K134, is known to be ubiquitinated during CMV infection and that Lamin B degradation is also needed for CMV infection makes it very hard to deduce anything about the specific role of the acetylation in this process. The authors propose that the role of the K134 acetylation is to compete for ubiquitination, thus preventing K134 ubiquitination and Lamin B degradation, but they offer no experimental support for this model. I am unsure how the authors can convincingly show a role for the K134 acetylation, since any mutation of K134 will also affect its potential ubiquitination. At the very least, the authors should test the role of the charge mimetic mutation (K134R) on CMV infection. If this mutant is also more resistant to CMV replication this would suggest that the K134Q mutant effect is likely due to its inability to be ubiquitinated, and not its acetylation-mimetic nature.

>Response: The reviewer brings up a nice point that we are very interested in, i.e., whether there is an interplay between acetylation and ubiquitination, and we have performed additional experiments to address this comment.

> However, first, to correct some of the misunderstood background information, Lamin B1 is not known to be ubiquitinated during HCMV infection. LMNB1 ubiquitination was only reported as part of a larger screen for ubiquitination modifications, and that study was not performed in the context of an infection. Also, the degradation of LMNB1 has not yet been studied during infection; the knowledge so far indicates that LMNA/C is disrupted via a phosphorylation-mediated mechanism. We only mentioned the LMNB1 K134 ubiquitination to put our finding in context, and not ignore a previously identified PTM at this site (although not yet functionally investigated). We initially only meant to include this hypothesis as one of the possibilities that could be considered. We apologize for not being clear, and we have now reworded this in the discussion of the manuscript (pages 20-21).

> To further address the reviewer's comment, we have now generated an R mutant for K134. For completeness, we also generated R mutants for the other two investigated sites, K109 and K483. The mutations of K109 and K483 did not significantly impact virus titers (new panels Fig. 4f-g). The K134R mutation had an intermediate phenotype, leading to a decrease in virus that was not as pronounced as for K134Q (new Fig. 4h). This impact on virus titer can derive from a number of factors, including a difference in the number of released viral capsids from the nucleus, a disruption in the assembly of viral particles, or a reduction in the overall infectivity of the produced particles. To start to address this, we also measured cell-associated virus. K134Q

expressing cells showed a significant reduction in cell-associated virus, while K134R had an intermediate phenotype, which was not significant when compared to cells expressing WT LMNB1. This suggests impaired capsid nuclear egress can in part contribute to the observed reduction in the virus titers for K134Q.

> In light of these new results, we now expanded our microscopy analyses of viral capsid nuclear egress to compare cells expressing WT, K134Q and K134R (Fig. 5a-d). Late in infection (120 hpi), slightly more viral capsids were accumulated in the nucleus in the acetyl-mimic mutant (K134Q) cells in comparison to both WT and K134R LMNB1 (Fig. 5b-c), suggesting an inhibition of nuclear capsid egress. Additional images of 120 hpi are shown as both single slice and maximum projection in Supplementary Fig. 9. In assessing the 3D volume of the pUL32-GFP particles inside the nucleus, we observed that K134Q cells displayed an accumulation of aggregated particles when compared to both WT and K134R cells (Fig. 5d). Altogether, these observations suggest that the phenotype is likely not driven by an acetylation-ubiquitination switch at K134, but rather a specific impact of acetylation (or the change in the charge of the side chain at that position) on the lamina.

> An interesting observation was that, at an intermediate time point (96 hpi), nuclear capsid egress seemed to progress more effectively in K134R cells (than in K134Q and WT LMNB1 cells) (Fig. 5b-c); this difference was only temporary, and capsid egress was similar in WT and K134R cells later in infection. However, this may further highlight the importance of this site in lamin regulation. Therefore, we further investigated the lamin morphology in the presence of these mutations at late time points of infection (96 and 120 hpi, Fig. 5e and Supplementary Fig. 10). The cells expressing WT or K134R LMNB1 displayed the known disruption of the nuclear periphery late in infection, with infoldings of the lamina and the formation of a kidney-shape nucleus. The curvature of this kidney shape is known to accommodate the viral assembly complex (as illustrated in Fig. 1a and shown Fig. 5a). In contrast, the cells expressing K134Q displayed a significant reduction in the disruption of the nuclear periphery, with less infoldings observed (Fig. 5e-f). Kidney-shaped nuclei were visible, but had a less pronounced aspect than the WT and K134R LMNB1 cells (Fig. 5e). Additional cells that display all the observed phenotypes are shown in Supplementary Fig 10. These results demonstrate that K134 acetylation inhibits the disruption of the nuclear lamina.

> The description of these new results has been now included in our manuscript (pages 15-17), and we have also expanded our discussion (pages 20-22).

b. It is not clear how the experiment in Fig. 4d was done. Did the authors infect cells expressing the mutant Lamins and stained them for IE1 directly, or did they collect viruses from the cells (and if so was this only supernatant or also cell-associated?) and then titrate them using the IE1 staining? There is obviously a major difference between the two possibilities.

>Response: We apologize that it was unclear how the IE1-based titer assay was conducted (previous Fig. 4d, current Fig. 4f-i). We have now updated the text and the methods to clarify the method (page 15 and page 39). In brief, we infected cells expressing the LMNB1 constructs and then collected the supernatant at 120 hpi. We then used that supernatant to infect a reporter plate of WT fibroblasts that was then stained for IE1. Additionally, based on the reviewer's comment, we have added panel Fig. 4i that now shows the assessment of cell-associated infectious virus in cells expressing LMNB1 WT, K134Q, or K134R. For this, cells expressing the LMNB1

constructs were infected with HCMV. At 120 hpi, the cells were collected and frozen. Upon thawing, the cells were sonicated to release the cell-associated virus. This released virus was then used to infect a reporter plate of WT fibroblasts, which were stained for IE1. This result further supports our data that the K134Q (acetyl-mimic) significantly impedes the formation of infectious viral particles within the cell (Fig. 4i), while the K134R (charge-mimic) presents an intermediate phenotype, with a non-significant change in the amount of cell-associated virus compared to WT.

c. In Fig 5b-c the authors quantify the relative abundance of CMV in the nucleus and cytoplasm. They find a relatively small effect (<20%) of the K134Q mutant on this ratio, and only at 120 hrs post infection. This small effect does not seem to explain the major effect of this mutation on viral titer (Fig. 4d – 200-fold difference). The authors should acknowledge this discrepancy and offer some interpretation. As it stands, I would think that the effect of the mutation is most likely at a step later than nuclear egress.

>Response: We agree with the reviewer's comment and interpretation. This is why we have now added the new investigation of the disruption of the nuclear periphery. As detailed above (in response to point 2a), when compared to WT or K134R LMNB1, the cells expressing K134Q displayed a significant reduction in the disruption of the nuclear periphery (Fig. 5f). The K134Q cells had significantly less LMNB1 infoldings (Fig. 5e-f), which supports the differences observed in capsid nuclear egress. However, in addition to this effect on capsid nuclear egress, the fact that LMNB1 acetylation inhibits nuclear periphery disruption may impact multiple aspects of the late stages of HCMV infection. In particular, it is also noteworthy that, although kidney-shaped nuclei were visible in K134Q cells, these were less pronounced than in the WT and K134R LMNB1 cells. The curvature of the nucleus is known to be necessary for the proper formation of the assembly complex, and the less pronounced kidney-shape in acetyl-mimic cells suggests that the stability of the resulting assembly complex could also be impaired. Consequently, the assembled viral particles may be partly defective and less infectious. Therefore, the overall impact of LMNB1 K134 acetylation on virus titer could represent a cumulative effect on capsid nuclear egress and virion assembly. We have now included a discussion on this (pages 16-17).

3. The role of pUL26 acetylation: The authors generate two mutant viruses, one that carries the acetylation-mimetic mutant, K203Q, and one that carries the charge-mimetic mutant, K203R. At MOI of 3 and 0.25, the K203Q mutant is less fit than the wild-type, while the K203R is not statistically different than wt. The authors conclude that: "These data suggest that acetylation inhibits the functions of pUL26 in supporting viral replication" (Line 378). However, if the acetylation played an inhibitory role, one would expect that the K203R, which is unable to go acetylation, would have a higher fitness than the wt virus. There is in fact a small increase in the K203R fitness at the lower MOI (0.25), which is not statistically different. I would advise the authors to repeat these experiment in much lower MOI, such as 0.01, in which case smaller fitness differences are likely to be more visible. In any case the authors should address this point, as currently their conclusion is not fully supported by the data they present.

>Response: We thank the reviewer for highlighting this difference in the K203R versus the WT.

As suggested, we have repeated the experiment at a much lower MOI and now included a new panel in Fig. 6 (Fig. 6h). As the reviewer predicted, at the lower MOI (MOI = 0.05) the K203R mutant has a statistically significant higher fitness than the WT and the K203Q mutant. This new result supports our conclusion that acetylation inhibits the function of pUL26. We have added a discussion of this new data to the results (page 19) and the discussion (page 25).

Minor concerns:

1. The manuscript requires some editing. For examples: the text references Fig S4 instead of S3 (Line 305), the supplementary files contains editing remarks.

>Response: We thank the reviewer for catching these errors. We have checked that the text refers to the correct figures and tables, and have removed editing remarks.

Reviewers' Comments:

Reviewer #1:

Remarks to the Author:

The authors have addressed all my concerns and I am happy to recommend publication in Nature Communications.

Reviewer #2:

Remarks to the Author:

All of my concerns and suggestions have now been comprehensively addressed. The manuscript reads very well and will be a valuable resource for the community. I suggest that the manuscript be published without the need for further changes.

Reviewer #3:

Remarks to the Author:

The authors have addressed all of my concerns thoroughly and elegantly. I recommend to accept the manuscript.

REVIEWERS' COMMENTS:

Reviewer #1 (Remarks to the Author):

The authors have addressed all my concerns and I am happy to recommend publication in Nature Communications.

Reviewer #2 (Remarks to the Author):

All of my concerns and suggestions have now been comprehensively addressed. The manuscript reads very well and will be a valuable resource for the community. I suggest that the manuscript be published without the need for further changes.

Reviewer #3 (Remarks to the Author):

The authors have addressed all of my concerns thoroughly and elegantly. I recommend to accept the manuscript.

>Response: We thank the reviewers for all the constructive suggestions and for indicating that all the questions were adequately addressed. We appreciate the careful and timely review process offered by the editor and reviewers.